# A large-scale view of marine heatwaves revealed by archetype analysis

Christopher C. Chapman [1,2] ✉, Didier P. Monselesan [1], James S. Risbey [1], Ming Feng [2,3] & Bernadette M. Sloyan [1,2]

Marine heatwaves can have disastrous impacts on ecosystems and marine industries. Given their potential consequences, it is important to understand how broad-scale climate variability influences the probability of localised extreme events. Here, we employ an advanced data-mining methodology, archetype analysis, to identify large scale patterns and teleconnections that lead to marine extremes in certain regions. This methodology is applied to the Australasian region, where it identifies instances of anomalous sea-surface temperatures, frequently associated with marine heatwaves, as well as the broadscale oceanic and atmospheric conditions associated with those extreme events. Additionally, we use archetype analysis to assess the ability of a low-resolution climate model to accurately represent the teleconnection patterns associated with extreme climate variability, and discuss the implications for the predictability of these impactful events.

In recent years the number of high-profile and devastating marine heatwaves have brought increased public awareness and scientific focus to these events[1,2]. The growing recognition of their impacts has resulted in an intense effort to understand the physical drivers of these phenomena, with the ultimate goal of improving their prediction and providing information to enable adaptation and mitigation measures[3–7]. While it is understood that, at a local level, marine heatwaves can be caused by anomalous ocean heat transport, enhanced surface heating from the atmosphere, or reduced vertical exchange (or combination of all three)[2], the link between the local drivers and the broader environmental conditions that favour their development is an area of active research. A key tenet of modern climatology is that conditions at one location may be influenced by remote drivers, often many thousands of kilometres away, through *teleconnections*[8]. There is a strong desire to understand the role played by teleconnections in marine extremes[1,2,9–11], as large-scale variability typically has longer timescales, is better represented in coarse-resolution climate models, and is hence more predictable than smaller-scale local processes[12–14].

To date, the majority of studies that investigate marine heatwaves and cold-spells focus on detailed case studies of events at a particular geographic region[10,15–19], although there are a number of studies investigating the connection between larger regions and remote drivers[9,11,20,21]. To link the local extremes with remote drivers, the general approach taken to date is to begin by defining extreme events at one or more distinct locations, then explore statistical or dynamical connections between those events and large-scale climate modes such as an El-Niño[9–11,18,19]. As the analysis proceeds from local to global scales, we will call this approach the 'inside-out'.

While this 'inside-out' approach has dramatically advanced the understanding of the characteristics and physical drivers of marine extremes, unambiguously separating local and remote influences is difficult due to the complex interconnection between components within the climate system. As 'inside-out' approaches employing fixed region heat budgets are necessarily limited to analysing the local drivers of marine heatwaves there is merit in considering large-scale dynamical frameworks that connect remote drivers to marine heatwave events[1].

In this study, we present an 'outside-in' methodology that directly identifies large-scale patterns associated with extreme sea-surface temperatures by employing a powerful data-mining methodology–Archetype Analysis (herein AA). AA seeks to represent a high-dimensional spatiotemporal dataset as a mixture of a smaller

[1]CSIRO Oceans and Atmosphere, Hobart Marine Laboratories, Castray Esplanade, Hobart, Tasmania, Australia. [2]Center for Southern Hemisphere Ocean Research, Hobart Marine Laboratories, Castray Esplanade, Hobart, Tasmania, Australia. [3]CSIRO Oceans and Atmosphere, Indian Ocean Marine Research Center, Crawley, Western Australia, Australia. ✉e-mail: chris.chapman@csiro.au

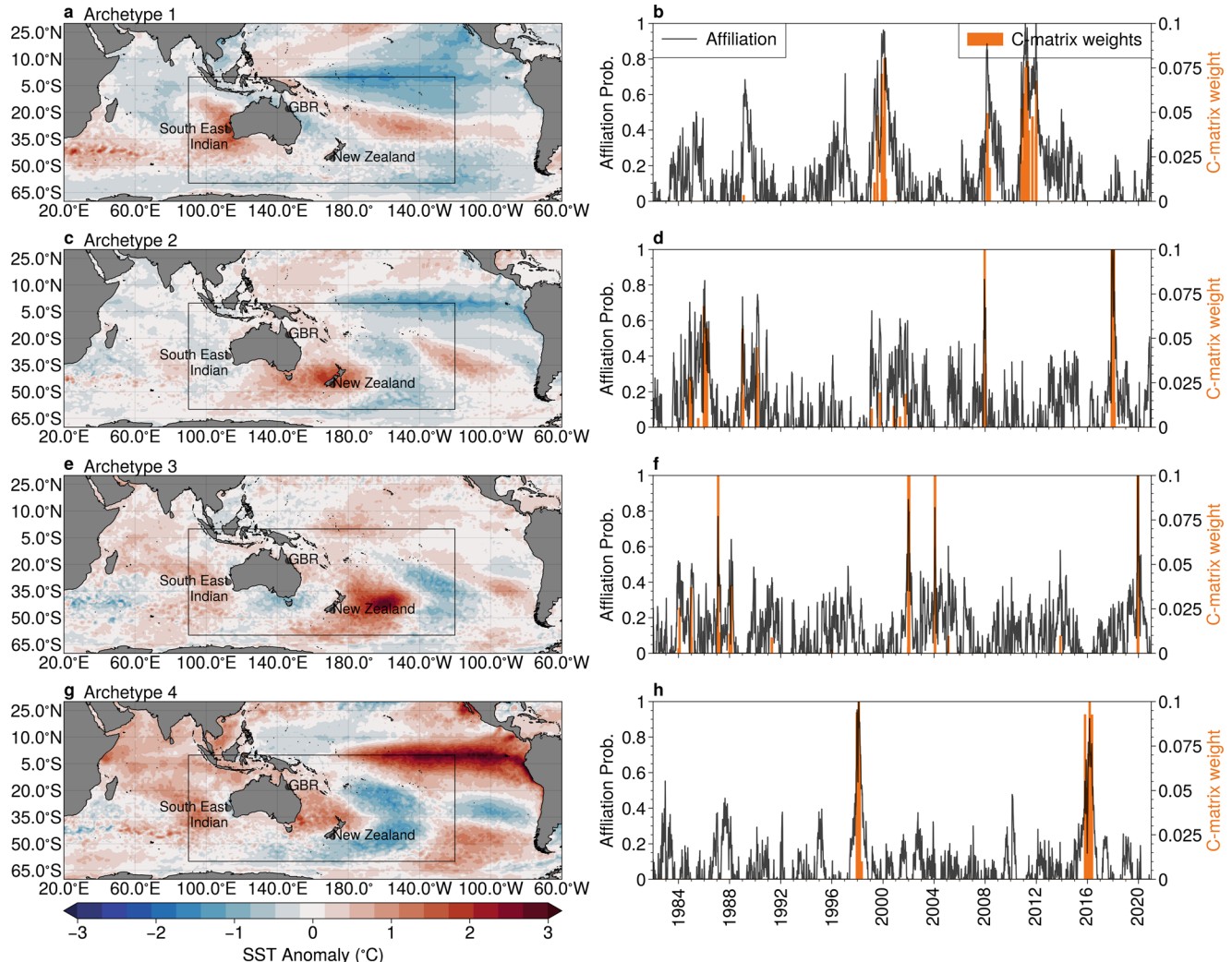

**Fig. 1 | Archetypal patterns and affiliation time series. a**, **c**, **e**, **g** Detrended sea-surface temperature (SST) anomalies for four of the archetypal patterns computed over the Australasian region (indicated by the black box), and **b**, **d**, **f**, **h** associated affiliation time-series (black solid line) and the weights applied to each time snapshot to form the archetypes. The four archetypes plotted here are selected based on their association with well-known marine heatwaves (locations indicated in the text). Maps created with Cartopy[71].

number of 'archetypal' spatial patterns along with a probabilistic time-series[22-25]. The archetypal patterns are themselves constructed as a weighted average of a small number of snapshots of the original dataset, $x$, that correspond to 'extreme' states[22,24,25]. An approximation of the original dataset, $\tilde{x}$, is given by:

$$x(\text{space, time}) \approx \tilde{x}(\text{space, time}) = \sum_{i}^{P} s_i(\text{time}) \, z_i(\text{space}) \quad (1)$$

where $z_i$ is the $i$th archetypal pattern, $P$ is the number of archetypes, and $s_i(\text{time})$ is the affiliation time series of the $i$th archetype, which can take values between 0 and 1. The archetypal patterns can be interpreted as extreme modes of variability, and the affiliation probability is the likelihood that one of these modes is expressed at any given time[24]. AA has been employed previously to identify the characteristics of extreme rainfall events[26] and long-lived atmospheric phenomena[27]. To the best of our knowledge, this work is the first application of AA to marine extremes.

Here, we apply AA to satellite-derived sea-surface temperature (SST) over the Australasian region to identify large-scale patterns that correspond to temperature extremes (i.e. marine heatwaves and marine cold spells) and show that AA unambiguously identifies

teleconnection patterns associated with extreme events. What is more, AA is able to reveal subtleties—for example distinguishing between 'classical' El-Niños and central Pacific (Modoki) El-Niños. Once the large-scale archetypal patterns have been obtained, we then investigate their impact on specific regions—hence 'outside-in'. Finally, we apply AA to assess the capacity of a modern climate model to represent the teleconnections associated with marine temperature extremes.

## Results

### Extreme climate modes in the Australasian regions

To begin our analysis, we apply AA to 39 years of SST anomalies over the southwestern Pacific and southeastern Indian Ocean basins (Fig. 1) (see Methods). After experimentation we chose a total of eight archetypes, of which four associated with marine heatwave conditions in Australasia are shown in Fig. 1 (the remainder are shown in Supplementary Fig. 1). Although the AA methodology is applied only to the Australasian domain (black box in Fig. 1) we plot the resulting archetypes by compositing over the southern Indo-Pacific to show the broad-scale SST patterns. The labelling of the archetypes is arbitrary.

In Fig. 1, we immediately recognise spatial patterns associated with classical and central Pacific (Modoki) La-Niña (archetypes #1 and

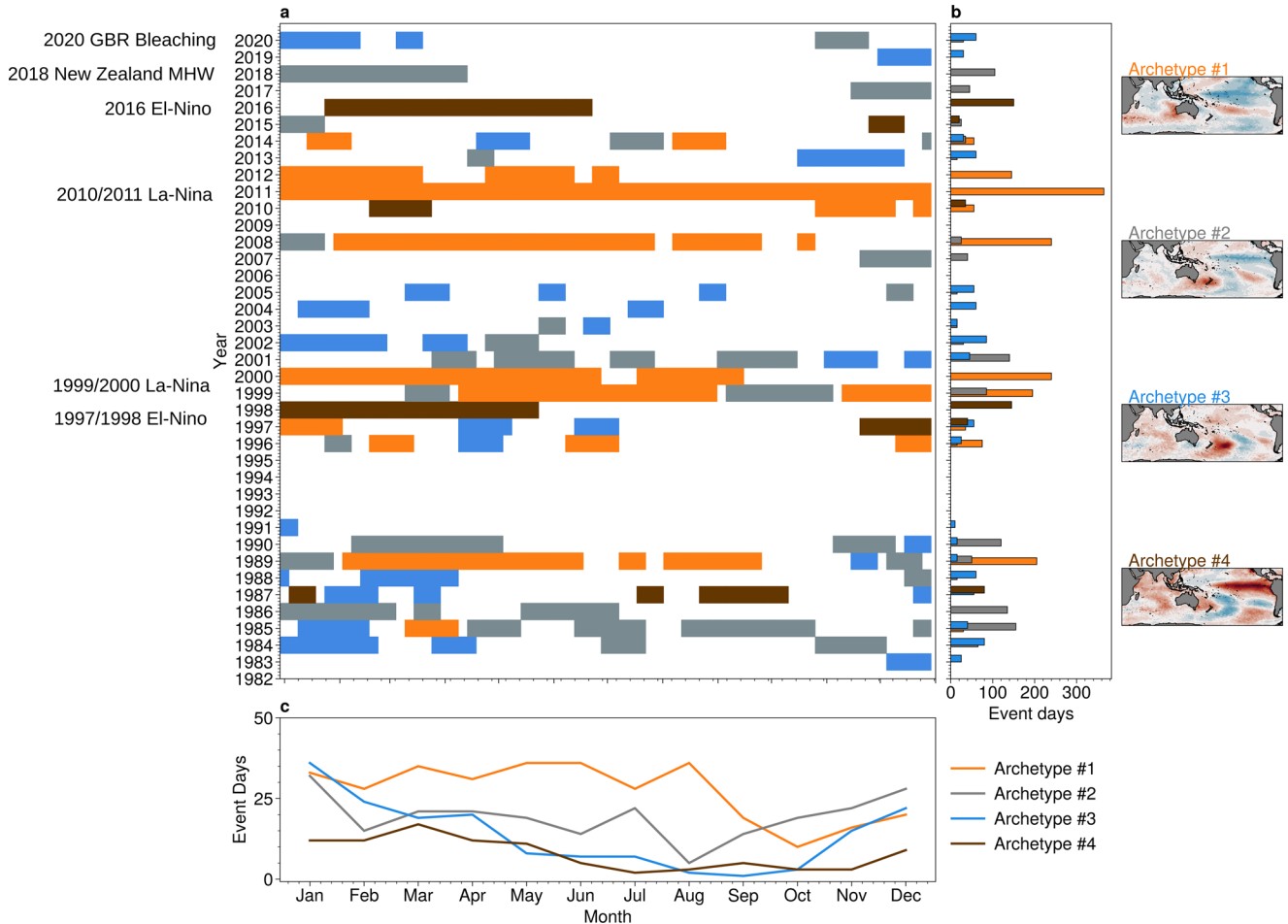

**Fig. 2 | Temporal regimes of archetypal patterns. a** Coloured blocks indicate periods where a particular archetype was dominant for at least 20 days. The *y*-axis indicates the year, while the *x*-axis indicates the calendar day-of-year. Blanked periods show days where no qualifying event was found; the total number of archetype event days for each archetype that occurs **b** for each year; and **c** for each month over the annual cycle. Maps to the right show spatial patterns of sea-surface temperature (SST) anomaly for each archetype. Significant climatological events are indicated by the text annotations. Maps created with Cartopy[71].

#2) and El-Niño patterns (archetypes #3 and #4). These inferences are supported by considering the affiliation time series (solid black line in the right-hand column of Fig. 1) and the temporal distribution of the weights that are used to construct the archetypal patterns (orange bars). For example, the weights used to construct the archetype #4 cluster around the years 1998 and 2016, which correspond to powerful El-Niño events[28]. In addition to the ENSO-like patterns in the equatorial Pacific, additional features are evident. For example, elevated SST anomalies are evident along the west coast of Australia (archetype #1, Fig. 1a), around New Zealand (archetype #2, Fig. 1c), and through the Great Barrier Reef region (archetypes #3 and #4, Fig. 1e, g). These patterns show the large-scale conditions that are likely during periods when the affiliation time series are close to 1, which may favour the development of marine heatwaves in certain regions.

Investigation of the affiliation time series associated with the 4 archetypes reveals periods of persistence and recurrence. We show this in further detail in Fig. 2a, where each coloured block corresponds to periods where a particular archetype is both dominant (affiliation probability >1/2), and persists for at least 20 days (Supplementary Fig. 2 shows this information for all 8 archetypes). A number of persistent regimes can be identified, such as a 16-month period from November 2010 until June 2012, when archetype #1 was strongly expressed, which manifested as an exceptionally strong La-Niña[29], or the 6-8 months in 1998 and 2016 of archetype #4 that corresponded to powerful El-Niños[28]. The inter-annual variability determined by summation of the number of event days for each dominant archetype in

each year, shown in Fig. 2b, reveals substantial year-to-year variability: particular regimes dominate in certain years while being completely absent in others. Seasonality is indicated in Fig. 2c, which shows the total event days for each month. For example, the archetypes that most clearly resemble El-Niño (archetypes #3 and #4) show a clear expression in summer months. Further information on the timescales associated with each archetypal pattern can be found in the supplementary material (see Supplementary Figs. 3–7).

The key result from the previous analysis is that AA reveals broad-scale, recurrent and occasionally persistent modes of variability. We now investigate the links between regional ocean temperatures and these extreme climate modes through a series of case studies.

### South-Eastern Indian Ocean marine heatwaves

In the 2010–2011 austral summer, the southeastern Indian Ocean basin was the location of one of the most intense and devastating known marine heatwaves, with temperatures of more than 3 °C higher than the climatological average[16,30–32]. We investigate the relationship between broad-scale extreme modes identified by AA and marine heatwaves in this region.

In Fig. 3a, we show the SST anomalies for the day of the peak intensity of the 2010–2011 extreme marine heatwave event (1st of March 2011) at a representative location (30˚S, 112.5˚E, indicated by the grey circle) as well as the of SST anomaly composite average for all marine heatwaves detected at that location at their peak intensity. Both the snapshot (Fig. 3a) and the composite average (Fig. 3b) show

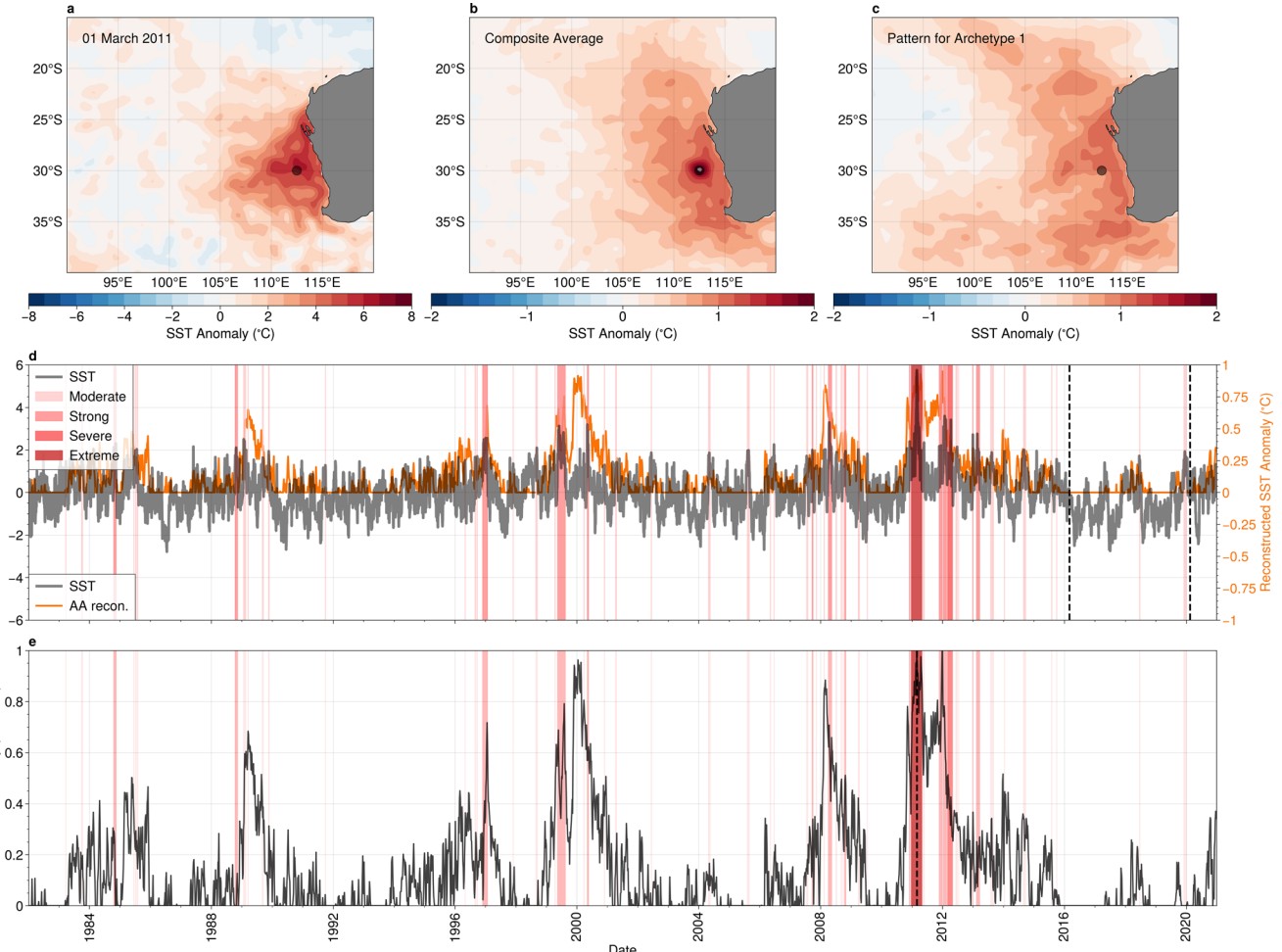

**Fig. 3 | The relationship between Marine Heat Waves and Archetype #1 in the Southeastern Indian Ocean. a** Snapshot of sea-surface temperature (SST) anomalies for the peak of the 2010–2011 extreme marine heatwave event, which occurred on the 1st of March, 2011; and **b** SST composite average for all marine heatwaves. Statistics are calculated at the representative location 30°S,112.5°E, indicated by the grey circle. **c** The SST anomalies for best matching archetypal pattern (archetype #1); **d** time-series of SST anomalies (black) and the reconstruction from archetype #1 (orange) at the representative location shown in panels **a**–**c**; **e** time-series of archetype affiliation probability for archetype 3. Coloured bands in panels **d**, **e** indicate marine heatwave occurrences, coded by the severity category described in Hobday et al.[32]. Maps created with Cartopy[71].

warm SSTs over a broad geographical range, from latitudes 20°S to 35°S, and eastward of longitude of 105°E, with the highest temperatures generally found closer to the continent. We identify a best matching archetype by comparing the spatial and temporal patterns shown in Fig. 3a, b with the archetypal patterns plotted in Fig. 1 (see Methods), in this case archetype #1 (Fig. 4c). The archetype has a similar distribution of warm SSTs over the region to that of the SST composite average north of 35°S, and a strong similarity to the SSTs associated with the 2011 event.

The temporal relationship between the regional marine heatwaves and the archetype can be revealed by examining the anomalous SST time series at our chosen representative location (Fig. 3d) and the affiliation time series for the best matching archetype (Fig. 3e). The SST anomaly time series shows low-frequency variation, with periods of above or below average temperatures that can persist for months or years. Marine heatwaves also exhibit low-frequency behaviour, with periods where several (occasionally high intensity) events cluster together, separated by longer periods with few, moderate-intensity events[33]. Periods with frequent marine heatwaves are unsurprisingly correlated with periods of higher than average SST.

The affiliation time series, shown in Fig. 3e, in highly correlated with periods of above average temperatures at our representative location. For example, the affiliation time-series on the 1st of March 2011, the date of the peak intensity of the extreme marine heatwave, is a maximum and approaches 1, indicating that the best matching archetype is the dominant regime during this period. The affiliation is generally high during periods of high SST and frequent marine heatwaves and low during periods with few marine heatwaves (e.g. 1990–1996 and 2001–2008). The reconstruction of the representative SST anomaly using a single archetype (shown as the orange curve in Fig. 3d) also deviates from 0 only during periods with frequent marine heatwaves.

We now employ AA to identify the teleconnection patterns that accompany the extreme events in this region. The mean spatial patterns for the satellite-derived SST anomalies associated with archetype #1 are shown for the Pacific and Indian ocean basins in Fig. 4a, while the surface air temperature and the mid-tropospheric atmospheric circulation (represented by the 500 hPa geopotential height and wind anomalies) are shown in Fig. 4b. Anomalously cool SSTs are found in the equatorial Pacific, with a temperature minimum found at -170°W, characteristic of the central Pacific (Modoki) La-Niña[34,35]. Investigation of the sub-surface temperature anomalies obtained from Argo floats along the equator associated with this archetype confirm this interpretation (Fig. 4c): a cool subsurface in the eastern Pacific reaching -300 m depth and surface expression in the central Pacific, co-

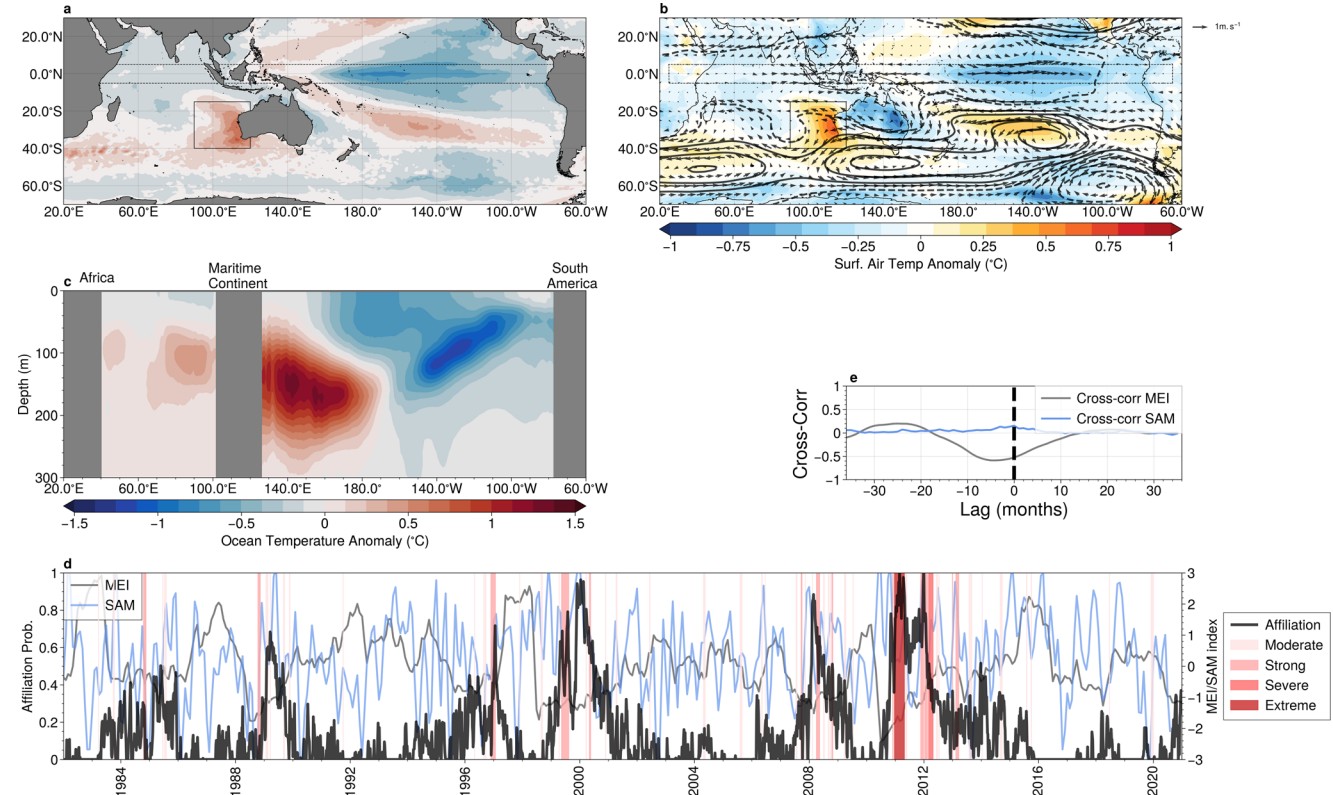

**Fig. 4 | Teleconnections Associated with Marine Heat Waves in the South-eastern Indian Ocean. a** Sea-surface temperature anomaly; **b** surface air temperature (colours) with anomalous mid-tropospheric (500 hPa) geopotential height (contour lines, contour interval 5m) and winds (vectors); and **c** subsurface ocean temperatures (averaged latitudinally between 5°S and 5°N within the region indicated by dashed lines in panels **a** and **b** associated with archetype #1. **d** The affiliation time series (solid black) together with the multivariate El-Niño index (MEI, grey) and the Marshall Southern Annular Mode (SAM) index (blue). Periods of marine heatwaves are indicated by red shading. **e** The lagged cross-correlation between the affiliation time series and the MEI (grey) and the Marshall SAM index (blue). Negative lags correspond to the MEI/SAM index leading the affiliation. Maps created with Cartopy[71].

occurring with a warm anomaly in the western Pacific. These conditions are known to initiate anomalously strong low-level trade winds in the tropical region, which in-turn result in an anomalous transport of warm water from the Pacific to the Indian basins via the Indonesian straits and an anomalous poleward heat transport[16,33,36], with the highest temperatures in the south-east Indian Ocean occurring ~3–5 months after the most intense negative anomalies in the equatorial Pacific.

The anomalous atmospheric circulation associated with this archetype (Fig. 3b) shows both local and remote anomalies. Locally, an anomalous cyclonic mid-tropospheric circulation directs airflow from the north and east, which are likely to contribute to the above average surface temperatures by bringing warmer continental and tropical air over the region[36,37]. Remotely, equatorial Pacific, mid-tropospheric winds show the characteristic divergence pattern associated with the central Pacific La-Niñas[34]. A ridge of high pressure extends across the Southern Ocean and anomalously strong eastward winds are found south of 55°S, characteristic of the positive phase of the Southern Annular Mode (SAM)[38].

To further illustrate the relationship between the archetype and the climate modes, we plot the affiliation time series in Fig. 4d together with the Multivariate El-Niño Southern Oscillation Index[39] (MEI) and the SAM index[40]. Significant anti-correlation between the affiliation time series and the MEI can be seen in Fig. 4d, confirmed by a lagged cross-correlation (Fig. 4e), which shows a negative cross-correlation coefficient of -0.5 with the MEI index leading the affiliation by ~6 months. We perform a similar lagged cross-correlation analysis against the Marshall SAM index, but find only weak correlation (maximum of 0.15) near zero lag.

Previous studies of the extreme 2010–2011 south-east Indian Ocean marine heatwave attribute ~2/3s of the excess warming to anomalous ocean heat transport, driven by remote conditions in the equatorial Pacific, while the remaining ~1/3 is due to enhanced surface heating driven by local atmospheric processes[16,36]. It is notable that the AA largely confirms these results, suggesting La-Niña influences in the equatorial Pacific, and local influences from a stationary mid-tropospheric cyclone. Our analysis indicates that the large-scale conditions responsible for the extreme 2010–2011 event are recurring and could form the basic ingredients of an extreme climate mode that strongly influences the south-east Indian ocean.

### South Pacific marine heatwaves near New Zealand

Our next case study concerns the South Pacific near New Zealand. This region suffered a severe category marine heatwave in the Austral summer of 2017–2018, which co-occurred with extreme land temperatures[41,42]. The impacts of this event were widespread, with the largest recorded annual loss of glacier ice mass in New Zealand's recorded history[41].

We plot the SST anomaly for the day of peak intensity of the 2017–2018 marine heatwave event at a representative location (here 45.9°S, 171°E, 5th December 2017) and the composite average of all events at this location in Fig. 5a, b. The spatial patterns in the single day snapshot and the composite average are very similar, albeit with different magnitudes, with warm SST centred near New Zealand's south island (approximate longitude 170°E, latitude 45°S), extending west into the Tasman Sea.

The pattern of the best matching archetype (archetype #2 of Fig. 1g, h), shown in Fig. 5c, has a remarkable visual similarity to those

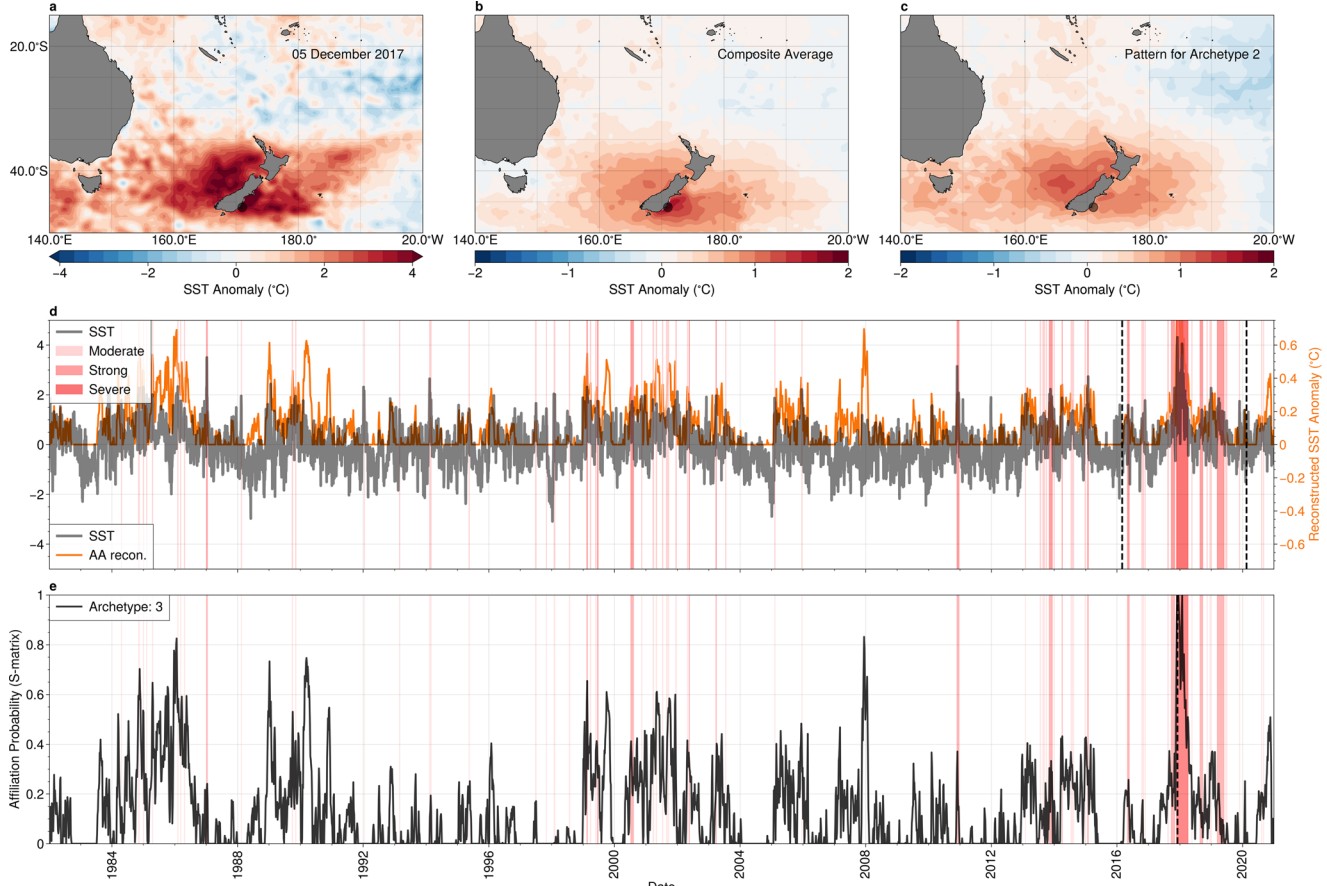

**Fig. 5 | The relationship between marine heat waves and archetype #2 in the Tasman Sea. a** Snapshot of sea-surface temperature (SST) anomalies for the peak of the 2017–2018 severe marine heatwave event, which occurred on the 5th of December, 2017; and **b** SST composite average for all marine heatwave detected at a representative location. Statistics are calculated at the representative location 45.9°S,171°E, indicated by the grey circle. **c** The SST anomalies for best matching archetypal (archetype #2); **d** time-series of SST anomalies (black) and the reconstruction from archetype #2 (orange) at the representative location shown in panels **a**-**c**; **e** time-series of archetype affiliation probability for archetype #2. Coloured bands in panels **d**, **e** indicate marine heatwave events. Maps created with Cartopy[71].

shown in Fig. 5a, b. The temporal evolution of the SST anomalies and the affiliation time series, shown in Fig. 5d, e, indicate that many, although not all, extreme events are captured by this archetype. As in the previous case study, marine heatwaves cluster, with a number of events occurring in a relatively short period of time, punctuated by longer periods with only a small number of isolated, weaker events. With only a single exception (between 1989 and 1991), the marine heatwave clusters occur during periods where the affiliation time series is persistently >0.5. Examples of these periods are 1984–1987, 1999–2004, 2005, 2013–2015; and 2018–2020, and the peak of the 2019–2020 severe marine heatwave (Fig. 5a) co-occurs with the absolute maximum of the affiliation time series. However, isolated marine heatwave events do occur during periods where archetype #2 is not strongly expressed.

As before, we examine the broad-scale SST (Fig. 6a); mid-tropospheric atmospheric circulation Fig. 6b; and equatorial sub-surface temperatures (Fig. 6c) associated with archetype #2. Concurrently with anomalously high SST centred on New Zealand (longitude ~170°E, latitude ~45°, indicated by the box in Fig. 6a–c), cooler SSTs are seen in the equatorial Pacific, extending from a longitude of 180° to South America. In contrast to the previous case study, the atmospheric circulation, surface air temperature (Fig. 6b) and the sub-surface ocean temperature (Fig. 6c) anomalies are weak in the equatorial Pacific. However, a strong blocking high-pressure system can be seen in the atmospheric field to the east of New Zealand. The anomalous atmospheric circulation directs warm air from the north, reduces

cloud cover over the region, and weak surface winds reduce the mixing of cooler, deep ocean waters with the surface, consistent with previous work[41,42].

The spatial patterns shown in Fig. 6a, b suggest that marine heatwaves around New Zealand are associated with classical La-Niña type patterns, as well as persistent atmospheric blocking high-pressure systems. However, a lagged cross-correlation shows only a weak correlation of the affiliation time series with the MEI (Fig. 6d and inset panel), with a peak correlation coefficient of −0.25 at a lag of zero, and we note that the magnitude of the ocean temperature anomalies in Fig. 6a, c are weak, which does not suggest a strong equatorial influence. Previous studies have also attributed marine heatwaves in the region to forcing associated with the Southern Annular Mode[41,42]. However, we find no significant correlation between the affiliation time series and the SAM index, and no SAM-like anomalous atmospheric circulation. The anomalous atmospheric circulation is reminiscent of the Pacific South-America (PSA) pattern[43], albeit displaced westward. The monthly PSA index is plotted together with the affiliation time series in Fig. 6d. However, we once again find a weak relationship, with a peak correlation coefficient of ~0.15.

Our analysis suggests that localised atmospheric blocking may be strongly linked with extreme and persistent marine heatwaves in the southern Tasman sea, and the role of broad-scale teleconnections is uncertain. The similarity of the archetypal SST patterns to that of the composite average and the clustering of events during periods when archetype #2 is strongly expressed also suggests these patterns are

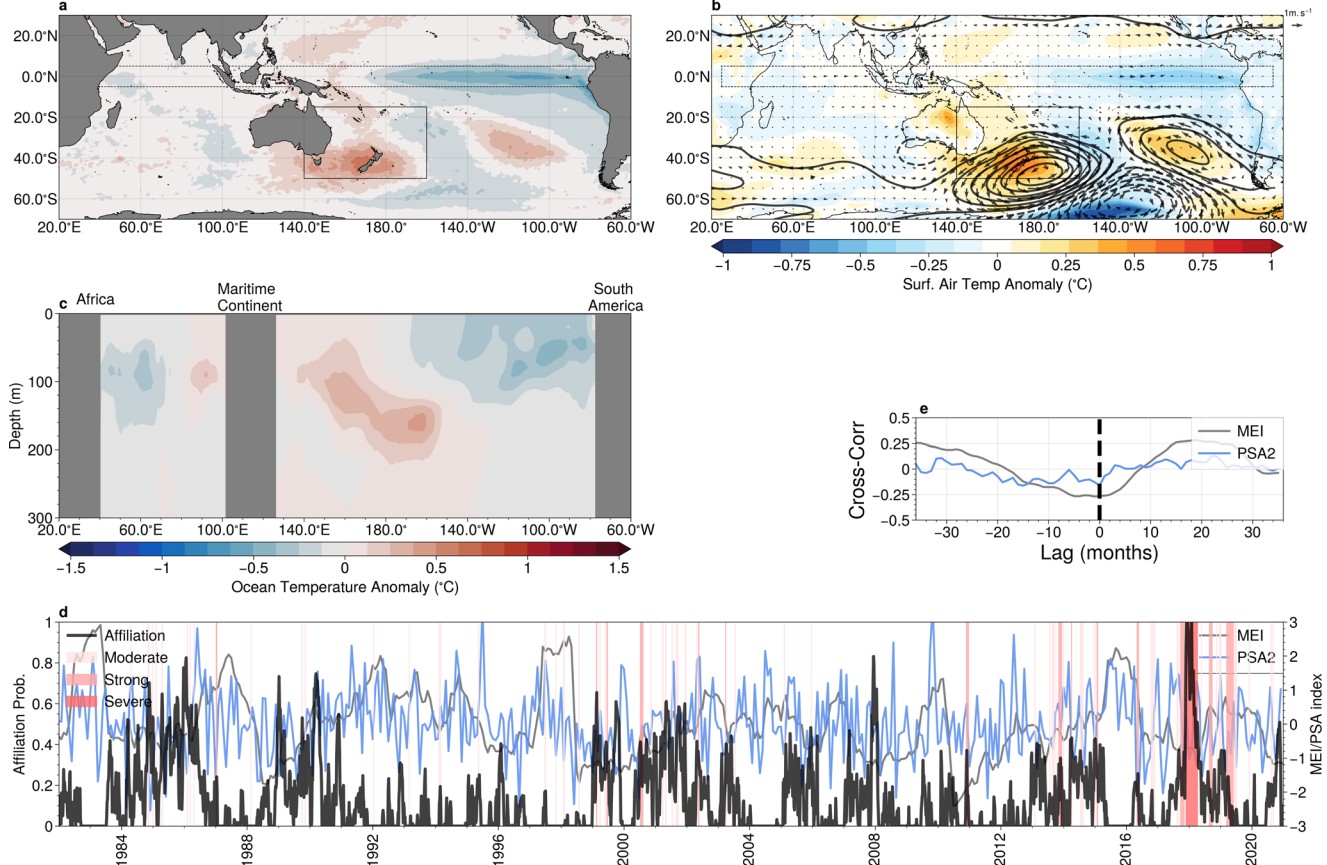

**Fig. 6 | Teleconnections associated with marine heat waves in the Tasman Sea.** **a** Sea-surface temperature anomaly; **b** surface air temperature (colours) with anomalous mid-troposphere (500 hPa) geopotential height (contour lines, contour interval 5 m) and winds (vectors); and **c** subsurface ocean temperatures (averaged latitudinaly between 5°S and 5°N within the region indicated by dashed lines in panels **a** and **b**), associated with archetype #2; **d** The affiliation time-series (solid black) together with the multivariate El-Niño index (MEI, grey), and the Pacific South America (PSA) index (blue). Periods of marine heatwaves are indicated by red shading. **e** The lagged cross-correlation between the affiliation time series and the MEI (grey) and the Pacific South America (PSA) pattern index (blue). Negative lags correspond to the MEI/PSA index leading the affiliation. Maps created with Cartopy[71].

reoccurring and associated with many (although not all) marine heatwaves in the region. While blocking highs are implicated in extreme marine heatwaves in numerous regions[17,44], there is currently no generally accepted theory that completely explains their dynamics[45]. Certain persistent atmospheric regimes, such as blocking, can be detected using AA[27], and future work could seek to integrate these analyses to improve understanding of the dynamics of these events.

## Coral sea and great barrier reef marine heatwaves

For our final case study, we investigate marine heatwaves in the Great Barrier Reef (GBR) to the north-east of Australia. Summertime marine heatwaves are associated with mass coral bleaching events, as high ocean temperatures are a necessary (but not sufficient) condition for coral bleaching[46,47]. The GBR suffered heat-induced mass bleaching events in 1998, 2002, 2006, followed by three events during the period 2016, 2017 and 2020[48].

In Fig. 7a, we show the composite average of anomalous SST for all summertime (December, January and February) marine heatwave events at a location representative of the "central" and "northern" regions of the GBR, as well as daily snapshots of anomalous SST for two marine heatwave events: March 2016 (Fig. 7b) and February 2020 (Fig. 7d), that were implicated in instances of mass coral bleaching. The composite average SST anomaly for all marine heatwave events shows that the highest SST anomalies lie close to the Australian coastline, tending cooler further offshore. A similar SST pattern can be seen in the daily snapshot of SST anomalies for the 2020 marine heatwave

event (Fig. 7b). The 2016 event, in contrast, shows elevated SSTs extending further north and more broadly over the Coral Sea.

Unlike in the previous case studies, we find that at least 2 archetypes, archetypes #3 (Fig. 7c) and #4 (Fig. 7e) are required to capture summertime marine heatwaves in the GBR region. These archetypal patterns show a similar spatial structure to the daily snap-shots at the day of the peak intensity of marine heatwaves detected in 2016 and 2020 (Fig. 7b, d). The necessity for two archetypes becomes clear when we investigate the relationship between the affiliation time series and the SST anomalies at the representative location (Fig. 7f, g). Marine heatwaves associated with major coral bleaching in 2006 and 2020 appear to co-occur with peaks in affiliation time series for Archetype #3, while Archetype #4 captures the conditions related to the 1998 and 2016 events. Neither archetype captures the 2017 coral bleaching event.

We now investigate large-scale patterns associated with summertime marine heatwaves in the GBR region. The anomalous SSTs (Fig. 8a), and surface air temperatures (Fig. 8b, colours) for archetype #4, which was strongly expressed during the severe 2016 coral bleaching event. The large-scale SST patterns show strong positive anomalies (of 1–1.5 °C) in the equatorial Pacific, characteristic of the mature phase of classical El-Niño. Concurrent warm surface temperatures occur throughout the Coral sea, northern Australia and the maritime continent. The affiliation time series for archetype #4, shown in Fig. 8d, shows a clear positive correlation with the MEI, with the lead-lag relationship suggesting that the archetypal pattern is most strongly expressed two to three months after the peak of temperatures in the

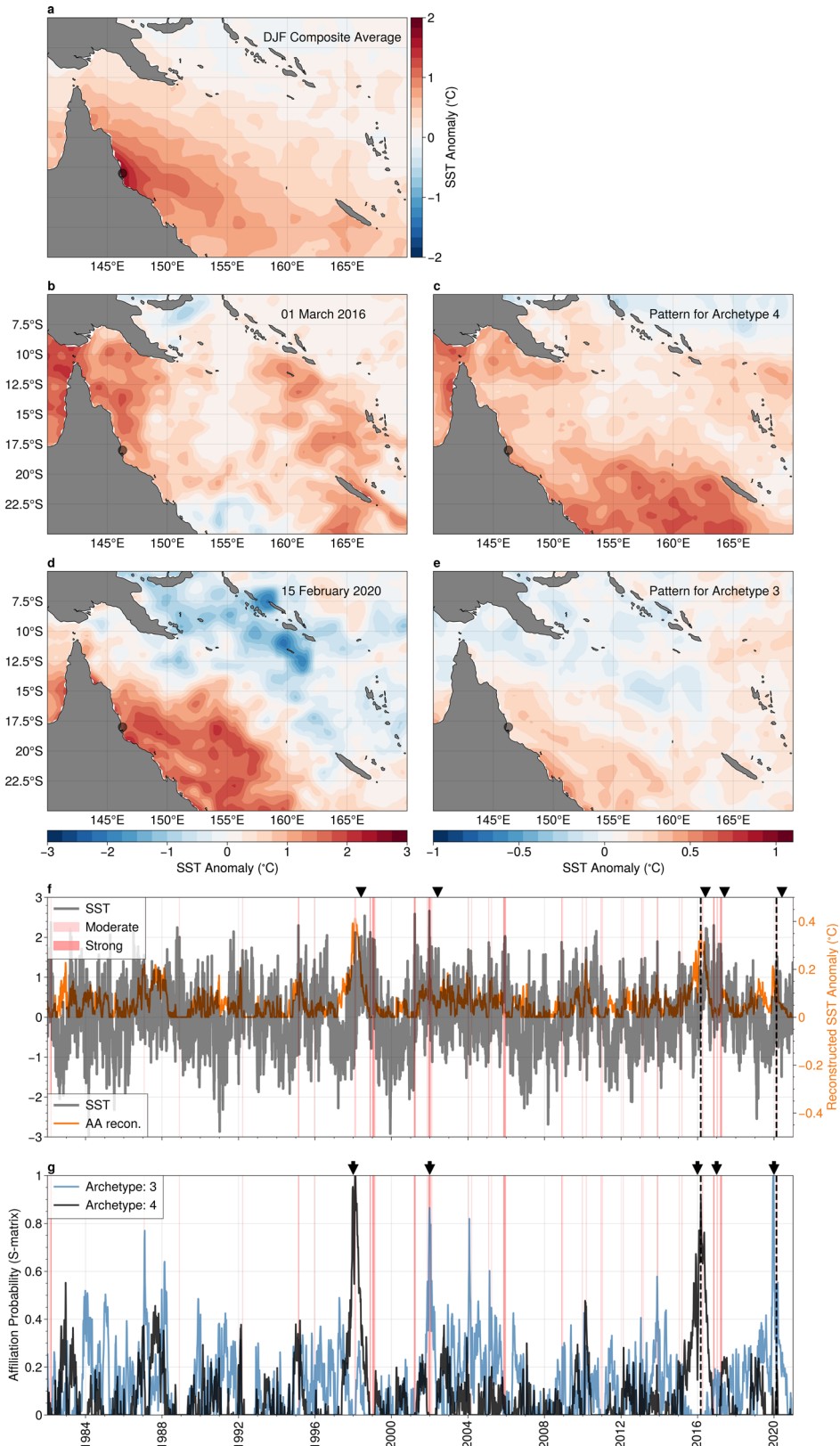

**Fig. 7 | The relationship between Marine Heat Waves and Archetypes #3 and #4 in the Great Barrier Reef region. a** Composite average of the sea-surface temperature (SST) anomaly for all summertime mearine heat waves at a representative location (18°S, 146.25°E), indicated by the grey circle; **c**, **d** daily SST anomaly snapshot for the peak of 2016 marine heatwave; **c** the SST anomalies for best matching archetypal pattern for the 2016 event (archetype #4); **d**, **e** as in **c**, **d** for the 2020 marine heat wave; **f** time-series of SST anomalies (black) and the reconstruction from archetype #3 and #4 (orange) at the representative location shown in panels **a**–**c**; **e** time-series of archetype affiliation probability for archetypes #3 and #4. Coloured bands in panels **d**, **e** indicate summertime (DJF) marine heatwave events, coded by the severity category. Black arrows in panel **g** indicate the occurrence of mass coral bleaching events. Maps created with Cartopy[71].

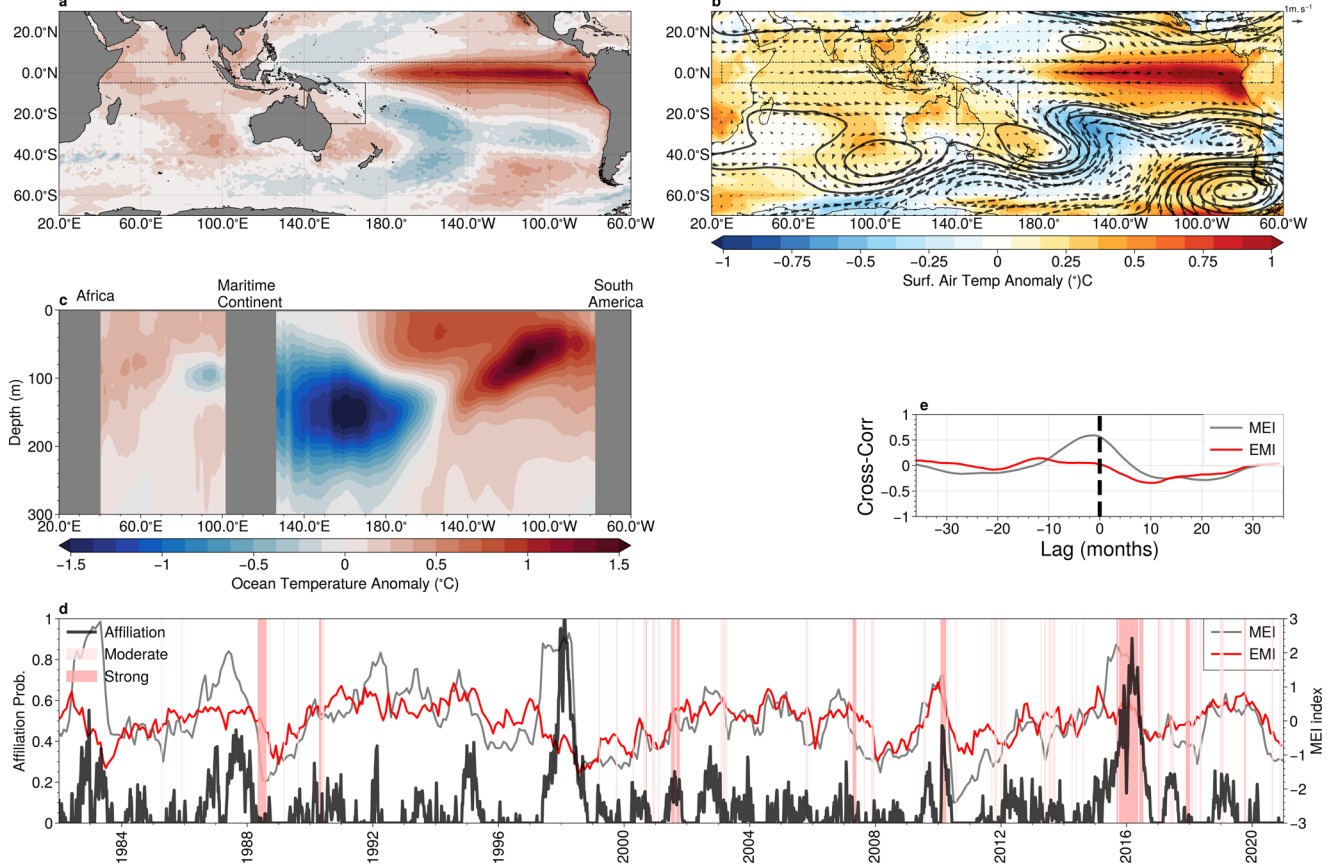

**Fig. 8 | Teleconnections associated with marine heat waves in the coral sea and great barrier reef. a** Sea-surface temperature (SST) anomaly; **b** surface air temperature (colours) with anomalous mid-tropospheric (500hPa) geopotential height (contour lines, contour interval 5m) and winds (vectors); and **c** subsurface ocean temperatures (averaged latitudinaly between 5°S and 5°N within the region indicated by dashed lines in panels **a** and **b**), associated with archetype #4; **d** The affiliation time-series (solid black) together with the multivariate El-Niño index (MEI, grey), and the Pacific South America (PSA) pattern index (blue). Periods of marine heatwaves are indicated by red shading. **e** The lagged cross-correlation between the affiliation time series and the MEI (grey) and the El-Niño Modoki Index (EMI) index (blue). Negative lags correspond to the MEI/EMI index leading the affiliation. Maps created with Cartopy[71].

tropical Pacific, consistent with the previous work[47,49]. The anomalous mid-troposphere circulation (Fig. 8b) shows weak anomalies over the GBR region, indicating little change in trade-wind conditions. However, surface air temperatures are elevated throughout the region, consistent with powerful El-Niño conditions[50]. Our results conflict somewhat with previous results that indicate local wind suppression as driving elevated SST over the GBR during El-Niños[49] and further investigation is merited.

Unlike with archetype #4 above, the broad-scale spatial patterns associated with archetype #3 (Fig. 9) show only weak ocean temperature anomalies (both surface and subsurface) in the central equatorial Pacific, and weak correlation with the ENSO Modoki index (maximum correlation coefficient of 0.25). The principal feature of the mid-tropospheric atmospheric circulation patterns associated with this archetype is a large blocking high to the east of New Zealand (centred at longitude 20°W, latitude 50°S), with an accompanying cyclonic circulation to the southeast of the Australia (centred at longitude 165°E, latitude 50°S). This cyclonic circulation directly opposes the westward trade winds that tend to dominate the summertime conditions in the GBR region. Reduced wind speeds can induce surface ocean warming through decreased evaporative cooling and inhibited upwelling of cooler, deep waters to the surface[51,52]. Investigation of the 2020 bleaching event indicated that reduced evaporative cooling of the ocean surface due to weak winds was the primary driver of this event, with increased solar heating playing a secondary role[4]. Our analysis broadly supports this interpretation by elucidating a potential remote driver of the suppressed trade winds, the large cyclonic circulation south-east of Australia.

## Additional case studies
In this study, we have focused on marine heatwaves that are efficiently described by AA. However, AA is capable of representing cold extremes as well. To illustrate this, a marine cold spell case study is included in the supplementary material (Supplementary Figs. 11 and 12). In addition, it is important to note that the power of AA lies in its ability to recognise extreme states over large spatial scales. As such, extremes at a regional scale driven by local processes may not be well captured by AA, which is shown in Supplementary Figs. 13–15, demonstrate two cases that are not well captured by AA due to the dominance of local influences.

## Evaluation of teleconnections associated with extremes in a climate model
Numerical ocean and climate models are employed both to predict distinct extreme events on timescales of days to weeks[3,4,7], for attribution studies of particular events[19], or for future climate projections[20,53]. However, as is well known, climate models are imperfect representations of reality, and the representation of extreme events in numerical models is sensitive to representation of physical processes and biases[20,54]. Climate models, particularly at the coarse resolution used for climate projections, typically do not capture the

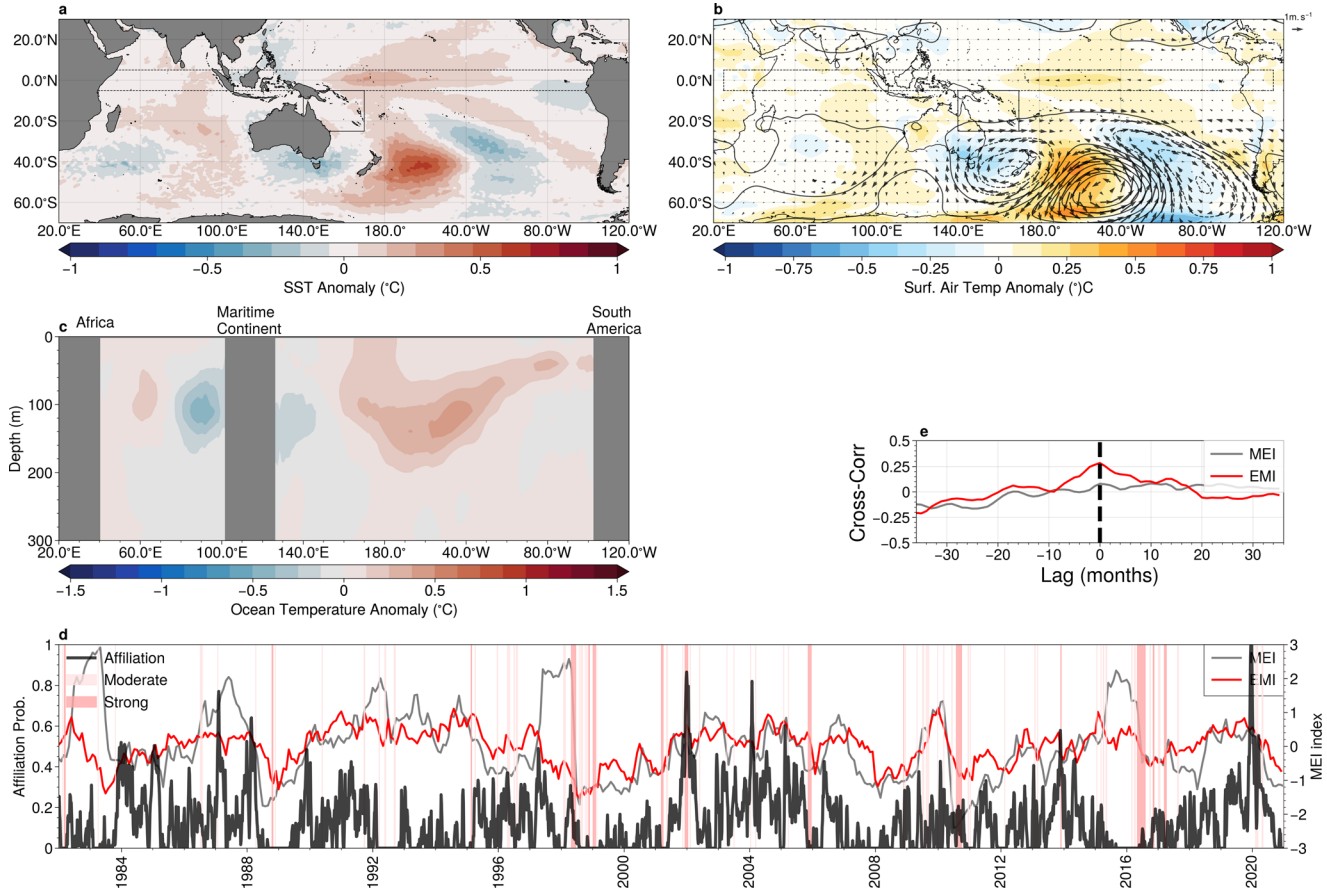

**Fig. 9 | Teleconnections Associated with Marine Heat Waves in the Coral Sea and Great Barrier Reef. a** Sea-surface temperature (SST) anomaly; **b** surface air temperature (colours) with anomalous mid-tropospheric (500 hPa) geopotential height (contour lines, contour interval 5m) and winds (vectors); and **c** subsurface ocean temperatures (averaged latitudinaly between 5°S and 5°N within the region indicated by dashed lines in panels **a** and **b**), associated with archetype #3; **d** The affiliation time-series (solid black) together with the multivariate El-Niño index (MEI, grey), and the Pacific South America (PSA) pattern index (blue). Periods of marine heatwaves are indicated by red shading. **e** The lagged cross-correlation between the affiliation time series and the MEI (grey) and the El-Niño Modoki Index (EMI) index (blue). Negative lags correspond to the MEI/EMI index leading the affiliation. Maps created with Cartopy[71].

tail of the temperature probability distribution, and produce 'extremes' that are not as intense or frequent as in reality[54,55].

However, if climate models are able to capture the broad-scale teleconnections associated with local extreme events, it may be less important that the model is incapable of representing the subtleties of those events at the local scale. A model may well approximate, for example, the teleconnection patterns associated with El-Niño, and hence an increased probability of extreme events in certain regions, even if the model does not capture the localised extreme events themselves. Down-scaling may improve the representation of the local extremes, but only in the case that the teleconnections are captured by the low-resolution model[55].

We employ AA to assess the capability of a coupled climate model to represent the extreme broad-scale patterns. We apply the technique to a long run the Australian Community Earth Systems Simulator-Decadal (ACCESS-D), with steady radiative forcing set at perpetual 1990 levels[56] (see methods). Eight archetypes are obtained from the final 39 years of detrended model SST anomalies (i.e., the same length as satellite SST observations), over a domain identical to the observational case studies. In Fig. 10 we show the four climate model archetypes most similar to those utilised in the previous case studies. For each archetypal pattern we show the large-scale SST anomaly (left column), the surface air-temperature anomaly, and anomalous mid-tropospheric circulation (centre) and the affiliation time series along with the C-matrix weights used to construct the archetypes (right).

For south-east Indian ocean (Fig. 10a–c) and Tasman Sea marine heatwaves, we note a strong similarity between the climate model archetypal patterns and those obtained from the observations (shown in Figs. 4 and 6). In the case of the Southeast Indian region, the model archetypal patterns show similar anomalous SSTs along the west Australian coastline and cool equatorial Pacific SSTs, reflective of La-Niña like conditions, although the model places the coolest SST anomalies further to the west than in the observations. The climate model also accurately reproduces the broad atmospheric circulation anomalies and surface air temperatures over the Australian continent (Fig. 10b). The model realistically simulates conditions to those identified in the New Zealand case study (Fig. 10d–f). In particular, a large atmospheric blocking high-pressure system is found in the region of highest SST anomalies near New Zealand, although we note that the high-pressure centre is shifted significantly to the south and east when compared with observations. In contrast to the observational case study, equatorial Pacific SSTs are anomalously warm.

However, when we consider teleconnections associated with the model's El-Niño like modes, (Fig. 10g–i and j–l), we find the equatorial Pacific SST anomalies are significantly to the west of those observed in the satellite SST, which results in anomalously cool SSTs in the GBR and Coral Seas, instead of warm conditions. AA reveals clearly how biases in the representation of the equatorial Pacific impact the model's ability to simulate important teleconnections to this region.

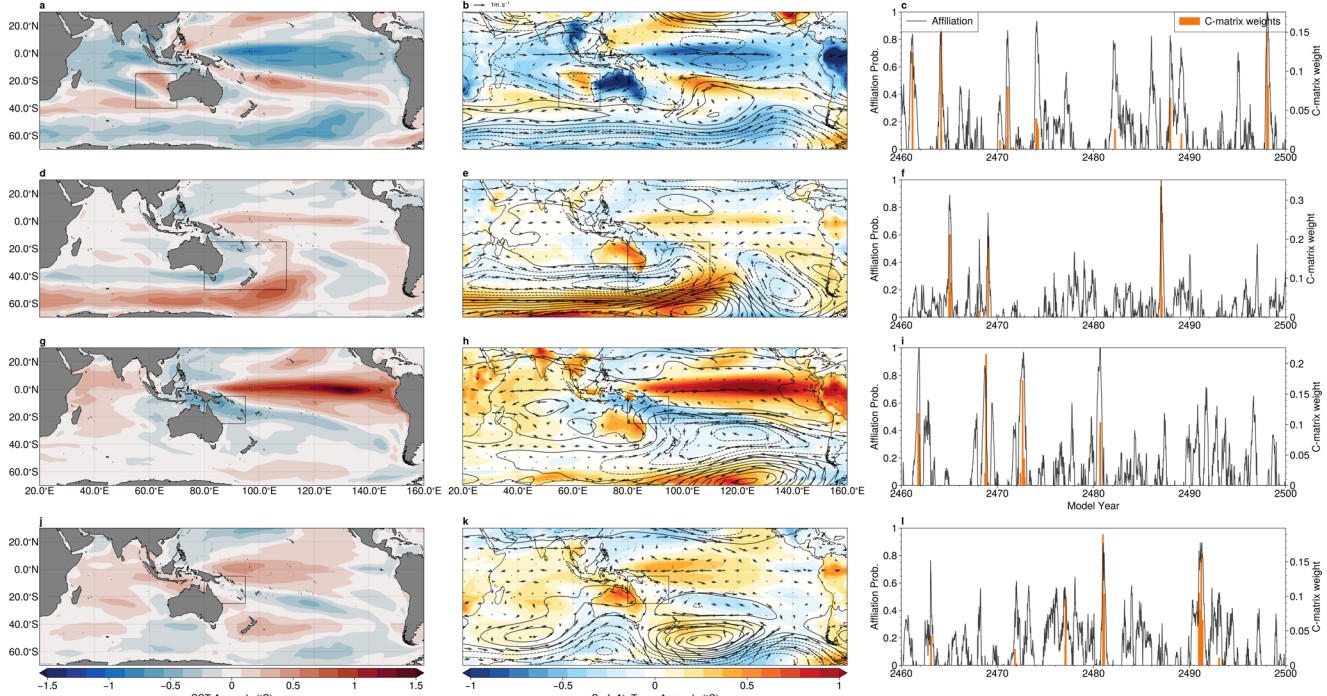

**Fig. 10 | Broad-scale patterns associated with marine extremes in a climate model. a, d, g, j:** anomalous sea-surface temperature; **b, e, h, k:** surface air temperature (colours), mid-tropospheric (500 hPa) atmospheric geopotential height anomaly (contour lines, contour interval 5 m), and wind (vectors). **c, f, i, l:** affiliation time-series (right) that correspond the archetypes that marine heatwaves in the **a–c** southeast Indian ocean, **d–f** southwestern Pacific near New Zealand; and **g–i** Great Barrier reef/Coral Sea region (indicated by box drawn in maps). Maps created with Cartopy[71].

## Discussion

In this work, we demonstrate that our 'outside-in' approach to characterizing extreme SST patterns using archetypal analysis (AA) is able to clearly identify the relationships between large-scale oceanic and atmospheric conditions and certain regional marine heatwaves. AA provides a minimal description of these extreme regional events, reducing the complex multi-faceted system to only one or two variables. The power of this approach is shown through several case studies, that identify not just the climate mode most likely to be associated with regional extreme SSTs (e.g. El-Niño or La-Niña), but also the atmospheric and oceanic teleconnection patterns, and the temporal relationships between those climate modes and the expression of the archetype, as well as the importance of the flavour or phase of ENSO (i.e. classical or central Pacific/Modoki).

Of the events studied here, both El-Niño and La-Niña conditions, in their various forms are identified as influences on marine heatwave occurrence in two of the three case studies (the southeast Indian Ocean and GBR/Coral Sea), and perhaps a secondary influence in the 3rd (Southwest Pacific/New Zealand events). However, our analysis has revealed that the type and phase of ENSO plays an exceedingly important role as well. For example, our investigation of the southeast Indian Ocean region has shown that the extreme climate mode is associated with the central Pacific (Modoki) phase of La-Niña. Applying AA to climate model output has also shown that subtle model biases, such as the position of warm equatorial SST in the model's simulation of El-Niño, can strongly influence the model's teleconnections. These results have implications for the prediction of marine heatwaves at timescales longer than a few weeks, or projection of marine heatwaves in future climate states.

It is important to note that while AA efficiently describes the large-scale patterns associated with extremes, it will not capture all individual events, particularly those driven by local processes. This should not be seen as a drawback, as this fact provides a mechanism for distinguishing between events driven by large-scale climate modes and those dominated by local processes. We have also not attempted to use AA to diagnose the distinct physical drivers of the events under study, and further work will prioritise blending process-based understanding with data-driven approaches. Although applied to marine heatwaves in this study, the approach can, in principle, be applied to a wide range of other physical phenomena, such as sea-level extremes or terrestrial heatwaves.

The approach presented in this study provides a viable and robust manner for linking large-scale variability and regional extreme events that, in turn, provides an improved understanding of the links between large-scale drivers and the local impacts.

## Methods

### Sea surface temperature data

In this study, we use a satellite-derived sea-surface temperature product as our primary dataset: version 2.1 of the National Oceanic and Atmospheric Administration's Optimum Interpolation SST Advanced Very High-Resolution Radiometer only product (NOAA OISST-AVHRR only) which has data for the period 1st January 1982 to 31st December 2020 (hence 39 years)[57,58] and anomalies are computed relative to this period. Data are provided at daily output frequency on a regular $0.25° × 0.25°$ regular latitude/longitude grid.

### Atmospheric reanalysis data

The atmospheric reanalysis used in this study is the Japanese 55-Year Reanalysis (JRA55)[59,60] provided on a $1.25° × 1.25°$ latitude/longitude grid. In our analysis, we use daily means of the original 6 hourly output, and restrict our attention to the period 1st January 1982 to 31st December 2020 (i.e. an identical period to that of SST data).

### Subsurface ocean temperature data

The subsurface temperature data employed is the optimally interpolated product developed at the Scripps Institute of Oceanography[61], which is based on profiles obtained by the international Argo

program[62]. This product provides estimates of ocean temperatures from the surface to 2000 db of depth, on a regular $1° \times 1°$ latitude/longitude grid. These data are only available from the year 2004 onward.

## Climate mode indices

In our regional case studies, we correlated affiliation time series for the best matching archetype with the various climate indices to illustrate the connection between the extreme modes identified by the AA and more familiar climate modes.

The Multivariate ENSO (MEI) Index (MEI) is a measure of the SST variability in the equatorial Pacific (30°S-30°N and 100°E-70°W) that uses principal component analysis to combine 5 oceanic and atmosphere variables (sea level pressure, sea surface temperature, zonal and meridional components of the surface wind, and outgoing longwave radiation) into a single index[39,63]. Large, positive values correspond to El-Niño conditions, while large negative values correspond to La-Niña conditions. Values are provided at a monthly frequency from December-January 1979 to present.

The Southern Annular Mode (SAM) Index reflects the position and strength of the westerly winds that blow over the Southern Ocean between latitudes of 40°S and 65°. The Marshall SAM index is constructed from sea-level atmospheric pressure observations taken at 12 weather stations on the Antarctic continent and some sub-Antarctic islands. Values are available at monthly intervals.

The Pacific South America (PSA)Index measures the strength of a large, quasi-stationary wave train extending from Australia to Argentina. It is defined in this study as the 2nd (PSA1) and 3rd (PSA2) principal component time series of the geopotential height anomalies at 500 hPa from the JRA55 reanalysis in the southern hemisphere. We note that the interpretation of the PSA is complicated by its definition using statistical properties (i.e. the principle components) and as opposed to a definition based on dynamics. In reality, the PSA may be composed of a superposition of travelling and stationary disturbances that can interact with each other[43], as the PSA1 and PSA2 modes, while being in approximate phase quadrature, have a relatively low coherence in the relevant frequency bands[27]. In this study, we use the generally accepted definition by convention, but the reader should bear in mind that the exact nature of the PSA is still a matter of some debate.

## Climate model

We use a 2500-year-long run variant of the GFDL Climate Model 2.1 (CM2.1)[64], used in older versions of the Australian Community Climate and Earth System Simulator (ACCESS). The model uses the same atmospheric, land, and sea ice components as CM2.1 (AM2, LM2, and SIS respectively) but uses the Modular Ocean Model (MOM4p1). The ocean model grid is the tripolar ACCESS-o grid[65] with a nominal grid spacing of 1° but with a finer latitudinal grid spacing in the tropics and the southern hemisphere high latitudes. There are 50 vertical levels, with 10 m grid spacing in the upper ocean, increasing to a maximum of 300 m. Subgrid processes for the ocean model are adopted from CM2.1, including neutral physics (Redi diffusivity and Gent-McWilliams skew diffusion), Brian-Lewis vertical mixing profile, Laplacian friction scheme and a K-profile parametrisation for the mixed layer calculation. The atmospheric model (AM2) has a grid spacing of 2° in latitude and 2.5° longitude, and 24 hybrid (sigma-pressure or terrain following pressure) vertical levels. Concentrations of atmospheric aerosols and radiative gases, and land cover are based on 1990 conditions. The model's ocean temperature and salinity fields are restored to World Ocean Atlas 2013 (WOA13) climatology at depths below 2000m, with a restoring time-scale of 1 year which improves the model's representation of the upper ocean stratification, at the expense of suppressing variability with multi-decadal time-scales, which is not the focus of this work. The model achieves approximate statistical

equilibrium after around 1500 years, with minimal drift in either temperature or salinity.

Although CM2.1 is an older climate model, used within the Coupled Model Inter-comparison Project phase 3 (CMIP3), we have opted to use it in this project due to its low numerical cost and relatively good performance in replicating the broad-scale variability over the Australian region.

## Archetype analysis

The analysis undertaken in this study employs Archetypal Analysis—an advanced data mining methodology that has been applied in fields ranging from marketing to astronomy. However, AA has only recently been applied to geophysics problems. Here we give a brief description of the AA problem and its implementation.

AA falls into a broad class of mathematical methods known as matrix factorisation. The goal of such methods is to represent a complex, high-dimensional dataset as the product of several, simpler and lower-dimensional datasets. In AA, for a given spatiotemporal dataset, $x(\mathbf{r}, t)$ represented as a data matrix $\mathbf{X} \in \mathbb{R}^{M \times T}$, where $T$ is the number of time observations and $M$ is the number of variables considered (i.e., number of grid-points in the SST dataset), we seek to find $P << M$ 'archetypal' states, $z$, that best represent the data:

$$x_{r,t} \approx \tilde{x}_{r,t} = \sum_{i}^{P} z_{r,i} s_{i,t} \quad i \in [1, P] \tag{2}$$

where the subscript $t$ refers to the time index, and the subscript $m$ to the spatial index. $s_{i,t}$ is the affiliation probability of the $i$th archetype, which is subject the constraints:

$$s_{i,t} \in [0, 1] \quad \text{and} \quad \sum_{i}^{P} s_{i,t} = 1. \tag{3}$$

The first of these constraints indicates that $s$ can only take values between 0 and 1, and the second indicates that, at any given time, the sum of the affiliations across all archetypes is equal to one. Matrices with this property are known as a left stochastic matrices. Mathematically, we say that $\tilde{x}$ is a convex combination of the archetypal patterns and corresponds only to an approximation of $x(\mathbf{r}, t)$. In AA, the archetypes themselves are written as required to reassemble the data. To enforce this, the archetypal patterns are written as a mixture of the data themselves:

$$z_{r,i} = \sum_{t}^{T} x_{r,t} c_{t,i} \quad i \in [1, P] \tag{4}$$

where $c_{t,j}$ are the mixture weights for archetype $j$, which have the constraints:

$$c_{t,j} \in [0, 1] \quad \text{and} \quad \sum_{t}^{T} c_{t,j} = 1. \tag{5}$$

Matrices with this property are known as left stochastic matrices. Like with the affiliation probability, the $c$ weights are constrained to take values between 0 and 1. The weights associated with the $i$th archetype, $c_{t,i}$ sum to 1 over all time steps. In the case that the number of archetypes is equal to the number of time-steps in the dataset $x(\mathbf{r}, t) = \tilde{x}$, a trivial solution where each archetype corresponds to a the field at a single time-step.

Combining Eqns. (2) and (4) gives:

$$x_{r,t} \approx \tilde{x}_{r,t} = \sum_{i}^{P} \sum_{j}^{T} x_{r,j} c_{j,i} s_{i,t} = \mathbf{XCS} \tag{6}$$

where we write the double summation as a matrix product between the original data matrix $\mathbf{X} \in \mathbb{R}^{M \times T}$, the C-matrix $\mathbf{C} \in \mathbb{R}_{\geq 0}^{T \times P}$, and the affiliation matrix $\mathbf{S} \in \mathbb{R}_{\geq 0}^{P \times T}$. The archetypal spatial patterns, such as those shown in Fig. 1, are given by:

$$\mathbf{Z} = \mathbf{XC} \in \mathbb{R}^{M \times P} \qquad (7)$$

The problem is now: for a given data matrix $\mathbf{X}$, can we find the $\mathbf{C}$ and $\mathbf{S}$ matrices that minimize:

$$\{\mathbf{S}, \mathbf{C}\} = \arg\min_{\mathbf{S}, \mathbf{C}} \| \mathbf{X} - \mathbf{XCS} \|_F \qquad (8)$$

where $\| \cdot \|_F$ is the *Froebenius norm*, defined as the square root of the sum of the squared absolute value of all matrix elements.

While the manipulations written above may seem esoteric, the AA decomposition has a relatively straightforward interpretation. Equation (7) states that the spatial archetypal patterns are simply an average of the original data weighted by the elements of the *C*-matrix of the original data, while Eq. (6) shows that the original data can be approximated by an average of the archetypal patterns weighted by the elements of the *S*-matrix.

Formally, the problem above can be shown to be equivalent to finding a discreet approximation to the convex hull of the dataset[22,24,66]. The convex hull is defined as the smallest convex 'envelope' of a dataset, and can be considered to be the boundary of a (potentially high dimensional) dataset. Since the convex hull of a dataset and the convex hull of its extreme points are identical, approximating the convex hull is equivalent to finding the extreme points (or corners) of the data underlying distribution.

As shown by Cutler & Breiman[22], the archetypal patterns are (approximately) located on the convex hull and are, therefore, approximations to the (high dimensional) extremes of the data. This astonishing result occurs due to the constraints imposed on the *S* and *C* matrices in Eqs. (3) and (5): that these matrices are non-negative and stochastic[66].

We note that native implementation of the AA algorithm as described by Eq. (6) is incapable of directly extracting temporal patterns, such as serial correlation or persistence, from data[25]. For example, a re-ordering of the time index of the data matrix, represented by the operation $\mathbf{X}' = \mathbf{RX}$ will result in reordered but otherwise identical affiliation and mixture weight matrices, given by $\mathbf{S}' = \mathbf{SR}^T$ and $\mathbf{C}' = \mathbf{RC}$. Although options for including temporal patterns directly into the AA procedure have been discussed[25], at present, it is only possible to extract through interrogation of the resultant affiliation time-series from the *S* matrix.

## Numerical implementation of archetype analysis

The minimization problem posed in Eq. (8) has no analytic solution for all but the simplest datasets and must be solved numerically in real-world applications. However, AA falls into a class of problems (non-negative matrix factorisation) that are known to be *NP-Hard*[67], which implies that, in general, only approximations to the 'true' solution can be obtained.

An increasing number of open-source AA algorithms are freely available and have been implemented for most major computing language in use today. Throughout this work, we rely on the MatLab implementation, PCHA, by Mørup & Hansen[68], whereby the optimization problem sketched in Eq. (8) is efficiently computed through a simple, but robust, projected gradient method.

In order to deal with the high dimensionality of geophysical fields, we apply a modification of AA, coined Reduced Space AA (RSAA), introduced in ref. 69 to reduce the spatial dimension of the problem and its computational burden. RSAA takes advantage of the invariance

of the Froebenius norm in Eq. (8) under unitary transformation:

$$\{\mathbf{S}, \mathbf{C}\} = \arg\min_{\mathbf{S}, \mathbf{C}} \| \mathbf{U}\boldsymbol{\Sigma}\mathbf{V} - \mathbf{U}\boldsymbol{\Sigma}\mathbf{VCS} \|_F = \arg\min_{\mathbf{S}, \mathbf{C}} \| \boldsymbol{\Sigma}\mathbf{V} - \boldsymbol{\Sigma}\mathbf{VCS} \|_F, \qquad (9)$$

where the spatial patterns or orthogonal empirical functions (EOFs) characterised by the unitary matrix $\mathbf{U} \in \mathbb{R}^{M \times R}$ can be factored out of the equation when a compact singular value decomposition (SVD) is applied to the data matrix $\mathbf{X} = \mathbf{U}\boldsymbol{\Sigma}\mathbf{V}$. The optimization is only performed on the scaled Principal Components (PCs), expressed as $\boldsymbol{\Sigma}V \in \mathbb{R}^{R \times T}$ in Eq. (9), with $\boldsymbol{\Sigma} \in \mathbb{R}_{>0}^{R \times R}$ the eigenvalue matrix where $R \leq \min\{M, T\}$ is the rank of the data matrix $\mathbf{X}$. To recover the archetypal patterns $\mathbf{Z}$, the solutions $\boldsymbol{\Sigma}\mathbf{VC}$ need to be left-multiplied by $\mathbf{U}$ such that $\mathbf{Z} = \mathbf{U}\boldsymbol{\Sigma}\mathbf{VC}$. Typically, RSAA uses a low-rank approximation of $\mathbf{X}$, $\tilde{\mathbf{X}} = \mathbf{U}\boldsymbol{\Sigma}'\mathbf{V}$, with $\boldsymbol{\Sigma}' \in \mathbb{R}_{>0}^{R' \times R'}$ and $R' << R$.

When applied to detrended OISST daily pentad (5 day averages) anomalies in the Australasian region (60 – 0° S, 90 – 240° E), the dimension reduction step allows a $M = 130349/R' = 840 \approx 155$-fold reduction in the number of variables, in our case grid points, the number of observations $T = 2849$ being left unchanged. The reduced rank $R' = 840$ is the number of retained PCs in the truncated SVD factorisation in Eq. (9). $R'$ corresponds to 95% of the total variance of $\mathbf{X}$. A similar approach and level of variance truncation are applied to the climate model data set.

Although the projection gradient algorithm used to solve Eq. (9) can be shown to converge to a solution for a suitable initialisation, there are however no guarantees that this solution is optimal given the *NP-Hard* character of the problem. An iterative procedure is required and achieved by resorting to multiple initialisations. Here, we combine one clustering and random-based initialisation strategies, whereby the data-driven 'FurthestSum' procedure advocated by[68] is compared to 999 random initialisations prescribed by[70] based on 'coreset' construction for AA. The optimal solution across 1000 trials is kept as the final result. For each individual trial, the projection gradient algorithm PCHA is considered converged when the relative sum of the square error stopping criterion reaches $10^{-8}$.

## Selection of the best matching archetype

Determination of the best matching archetype for the case studies presented is performed manually using semi-objective criteria. However, there is some subjectivity in the choice the best matching archetype and a truly objective method of determining would depend on the problem at hand.

First, the spatial SST pattern of each archetype is assessed at each representative location by linearly interpolating of the fields shown in Fig. 1. Only those archetypes with a strong expression at each representative location were considered. Then, the spatial patterns of the archetype are examined for similarity with the composite average of all marine heatwave events detected at that location (as in Figs. 3, 5 and 7), and the affiliation time series for that archetype examined for its similarity with the SST anomalies and the temporal distribution of marine heatwave events.

In the first two case studies presented here (as well as those in the supplementary material) the best matching archetype was relatively obvious based on spatial patterns alone. However, the Coral Sea case study required some care in selecting archetypes, as only summertime (December, January, February) events were considered (as these events lead to coral bleaching) and no one archetype was consistently consistently associated with summertime marine heatwaves. As such, we selected the only two archetypes that had positive expressions at the representative location during the appropriate season. Further detail is included in the supplementary material.

**Forming composite fields using affiliation (S-Matrix) and archetype weights (C-Matrix)**

Once an affiliation time-series has been computed, it can be applied as a weight to form clusters or composites in order to identify the climatic states associated with any particular archetype. For example, in this study, we have extracted atmospheric and sub-surface ocean patterns associated with the extreme states identified in the SST by the AA in order to demonstrate the remote teleconnections that may influence the regional extremes. This utility arises from the interpretation of the affiliation time-series as the probability at time $t$ that the data is associated with the $i$th archetypal pattern $z_i$[24]:

$$s_{i,t} = Pr(z_i | x_t). \tag{10}$$

As such, the affiliation can be used to associate any dataset with the $i$th archetypal state.

To derive the spatial fields of a supplemental dataset (for example, atmospheric geopotential height at 500 hPa) $y = y(\text{space, time}) = y_{m,t}$ associated with the $i$th archetype, we simply compute the temporal average of $y$ weighted by $s_{i,t}$:

$$\bar{y}_{m,i} = \frac{\sum_t^T y_{m,t} s_{i,t}}{\sum_t^T s_{i,t}} = \frac{\mathbf{YS}^T}{\sum_t^T s_{i,t}}. \tag{11}$$

In this study, AA is applied to an SST dataset with a temporal period from 1st January 1982 to 31 December 2020, at 5-day output frequency. As such, the affiliation time series spans an identical time period with identify output frequency. However, the atmospheric reanalysis spans a longer time period (1958-present) with daily output frequency, while the Argo-derived sub-surface temperature dataset spans a shorter time period (2004-present) at monthly output frequency. In order to apply Eq. (11) to these datasets, the JRA55 fields are first down-sampled to a 5-day output frequency by low-pass filtering the data using a standard box-car filter with a cut-off period of 1/5 days, then sub-sampled to 5-day output frequency, truncated to the same temporal period as the SST data, which allows direct application of (11). In the case of the sub-surface temperature dataset, the affiliation time series is down-sampled and truncated to match that of the Argo product.

A similar procedure using the archetype weights (C-matrix) may also be enacted. However, since $\sum_t^T c_{t,i} = 1$, the weighted average is simply:

$$\bar{y}_{m,i} = \sum_t^T y_{m,t} c_{t,i} = \mathbf{YC} \tag{12}$$

At any given time step in the data, $t$, the value of the affiliation time-series for archetype $p$, written $S_{p,t}$, expresses an estimate of the strength of the archetype's expression at that timestep with 0 lag. However, by shifting the time index of the affiliation probability with respect to the data to form lagged composites, which can be used to investigate the temporal propagation of certain spatial features and (equivalently) their associated timescales. We have alluded to the temporal evolution in each of the case studies, and present the results of lagged composite analysis in the supplementary material. Maps of the lagged composites for a discreet number of lags are shown in Supplementary Figs. 3–6, while point values for all lags are shown in Supplementary Fig. 7.

**Determination of the statistical significance of the composite fields**

At present, the question of determining the significance level of archetypal patterns is unresolved. As such, in this paper, the statistical significance of the composite fields formed from the weighted averages is assessed by a simple, brute force Monte-Carlo technique. To begin, we generate synthetic $S$ and $C$ matrices by populating each element with a random number drawn from a uniform distribution between 0 and 1. The rows or columns are then appropriately normalised to apply the constraints in Eqs. (3) or (5). We then form composite average fields using these synthetic matrices following Eq. (11) (when testing the significance of the composites formed with the affiliation time series) or Eq. (12) (for testing the significance of composites formed using the C-matrix). The procedure is then repeated 1000 times and the 95% and 5% percentile computed. The spatial patterns obtained from the AA are then tested against these synthetic composites: a single pixel is considered 'significant' (with 95% confidence level) if its value is less than the 5th percentile or greater than the 95th percentile.

Archetypal patterns and the associated composite averages are almost everywhere significant, as might be expected from a methodology that specifically extracts patterns associated with extreme states. This information is shown in Supplementary Figs. 8–10.

**Identification of marine heat wave/marine cold spell events**

The definition of Marine Heat Waves (and Marine Cold Spells as discussed in the supplementary material) follows that of Hobday et al. 2018[32] with a slight modification: a MHW is detected if the SST at a particular location exceeds the 90th percentile for a duration of at least 10 days (as opposed to the standard definition of 5 days). The temperature may briefly drop below the required thresholds for a period not exceeding 2 days and still be declared an extreme event. We have imposed a slightly more strict criteria on the persistence of events in order to eliminate multiple short duration, moderate intensity events that occur in near-coastal regions that appear to be more a response to high-frequency noise in the SST than modulated low-frequency variability.

## Data availability

We have made use of publicly available data only; no new data were generated as a result of this study. NOAA-OISST sea-surface temperature data is obtained from the National Oceanic and Atmospheric Administration's National Centers for Environmental Information (NOAA-NCEI), from (10.7289/V5SQ8XB5). The Japanese 55-year reanalysis project (JRA-55) are obtained from the National Center for Atmospheric Research (NCAR) Research Data Archive: https://doi.org/10.5065/D6HH6H41. The gridded subsurface ocean temperature dataset is downloaded from https://sio-argo.ucsd.edu/RG_Climatology.html. The Argo temperature profiles themselves were collected and made freely available by the International Argo Program and the national programs form the part of the Global Ocean Observing System (http://www.argo.ucsd.edu, http://argo.jcommops.org) Multivariate ENSO Index (MEI) v2 time series were published by the National Oceanic and Atmospheric Administration's Physical Science Laboratory and obtained from https://psl.noaa.gov/enso/mei/. The Marshall Southern Annular Mode (SAM) index These were published by the British Antarctic Survey and obtained from https://legacy.bas.ac.uk/met/gjma/sam.html.Map backgrounds were made with Natural Earth, free vector and raster map data (naturalearthdata.com).

## Code availability

Source code used for the generation of archetypes, written in the Matlab language, is freely available from the website of its author Morton Mørup: http://www.mortenmorup.dk/MMhomepageUpdated_files/Page327.htm. Source code for the climate model used in the study can be obtained from the National Oceanic and Atmospheric Administration (NOAA) Geophysical Fluid Dynamics Laboratory: https://data1.gfdl.noaa.gov/CM2.X/. Code for the definition of marine heat-waves following Hobday et al.[32] is available from https://github.com/ecjoliver/marineHeatWaves.

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

## Acknowledgements
The authors gratefully acknowledge financial support from the Center for Southern Hemisphere Ocean Research (CSHOR) project and from the Australian Climate Service. C.C., D.P.M., and J.R. received support from the Australian National Environmental Science Program. This research was undertaken with the assistance of resources and services from the National Computational Infrastructure (NCI), which is supported by the Australian Government.

## Author contributions
C.C. and D.P.M. conceived the study. D.P.M. was responsible for performing the archetype analysis. C.C. and D.P.M. performed the analysis of the results, with input from J.S.R., M.F. and B.M.S. C.C. was responsible for running the climate model experiments. C.C. lead the drafting of the manuscript, with inputs from D.P.M., J.S.R., M.F. and B.M.S.

## Competing interests
The authors declare no competing interests.
