## [Peer Review File · Nature Communications]

A Large-Scale View of Marine Heatwaves Revealed by Archetype AnalysisREVIEWER COMMENTS

Reviewer #1 (Remarks to the Author):

This manuscript describes application of a novel method, Archetype Analysis, to determine the role of large scale Ocean-Atmosphere patterns in the structure of Marine Heatwaves around Australia. The work is novel, the analysis is sound, and the manuscript should be accepted for publication subject to minor revision.

The figures are very good and support the points made in the text. I have only one major point. Some discussion, on the order of 3-4 paragraphs, should be added to the Discussion at the end regarding the application of AA in light of the temporal nature of teleconnections. It appears that AA works at establishing the connection between large scale Ocean-Atmosphere patterns at zero lag to the regional Marine Heatwave structure. As described in the text in lines 212-213 there are propagation time scales to consider. The lagged correlation relating to the climate indices was useful. It would be good to have a more general discussion of how AA relates to lags. I think it implies zero lag only, but it would be good to state this explicitly in the discussion. I only make this point as the AA methodology appears very promising and I expect this to be used in many other geographic region by less skilled researchers, and a clear directive in the discussion would be useful to future efforts. My other comments are minor and are listed below. A further check of spelling and grammar would be helpful, not sure I caught everything.

This was a very well-organized paper with important points and I strongly urge publication after adding the temporal points in the discussion and cleaning up the spelling and grammar suggested below.

Line 129. What is dominant in quantitative terms? > 0.5? Define.

Line 139 el Nino are > el Nino

Line 151 average was the > average. It was the (split into two sentences)

Line 153 that 3 C > than 3 C

Line 223 Pacific the surface air > Pacific surface air

Line 266 remove in Figs to both the

Fig. 5 caption marine heatwaves occurrences > marine heatwave occurrences

Line 329 studys > study

Line 356 run on sentence split into two

Line 367 shows > show

Line 369 temperatures through > temperatures occur through

Line 378 remove to have

Line 379 month > months

Line 387 near GBR > near the GBR

Fig. 7 caption 2nd line from bottom marine heatwaves occurrences > marine heatwave occurrences

Line 402 scale > scales

Line 453 closey > closely

Line 458 with extreme the > with extreme

Line 463 insteade > instead

Line 496 Why However? Doesn't this point just agree with the first point in the first sentence?

Line 503 remove Extremes

Line 584 we > We

Line 665 as an > an

Line 783 that > than

Reviewer #2 (Remarks to the Author):

Comments on "Large-Scale Drivers of Marine Heatwaves Revealed by Archetype Analysis"

In this paper, the authors use a novel data mining method – Archetype Analysis (AA) to investigate the large-scale drivers of Marine Heatwaves (MHWs) near Australasian waters. By applying AA to sea surface temperature from 1982-2020, they identified four archetypal patterns associated with MHWs in the domain of interest. They further reconstruct large-scale atmospheric and oceanic fields related to certain archetypal patterns and explore the drivers of several MHW events. This is a well-written paper introducing a new method to the rapidly growing literature on Marine Heatwaves. The introduction of the methods is clear and the descriptions of the figures are adequate. Overall, I recommend the publication of this work. I do, however, have comments on the discussion of the processes and mechanisms about the drivers of the events. My impression is that this work is more like 'proof-of-concept' type of study, rather than revealing the actual drivers of the heatwaves. The description of the mechanisms are mostly brief, vague and sometimes speculative, which does not provide much in-depth knowledge of the driving processes. Since the title of this manuscript contains "Drivers", I suggest that the authors expand the explanations and descriptions of the mechanisms driving the events discussed in the manuscript. Some examples can be found below.

Page 10, line 211: "The thermocline anomalies in the western Pacific are known to propagate via oceanic teleconnection to instigate the southeast Indian Ocean marine heatwave." The authors could have explained more how the oceanic teleconnection works, instead of just inserting a reference.

Page 10, line 243: "The dominant driver is found to be central Pacific La Niña, with the SAM potential secondary driver." Again, how does the process work? Also, be careful about whether or not La Nina (SST anomaly) is a driver or response of the actual driver. Similarly, how does SAM (wind variability) drive the SST anomaly in the domain of interest needs more than visual inspection of the patterns.

Page 11, 294: "The spatial patterns shown in Figs. 6a,b suggest that marine heatwaves around New Zealand are associated with classical La Niña type patterns, as well persistent atmospheric blocking high pressure systems." This sentence seems to suggest that every heatwave around New Zealand is associated with La Nina and blocking. Is this the case?

Page 14, line 318: "The analysis conducted here suggests that although both classical

La-Niña and PSA climate modes play a role on driving marine heatwaves around New Zealand, individually each of these climate drivers has only a weak influence, with the dominant role being played by localised atmospheric blocking high pressure systems.” Again, description of the driving process is inadequate.

Page 15, line 349: “Unlike in the previous case studies, we find that at least 2 archetypes, archetypes #3 (Fig 7c) and #4 (Fig. 7e) are required to ...”. The determination of which archetypes explain the heatwave seems arbitrary. Is there a more quantitative way of selecting the archetypes?

Page 19, line 393: “While archetype #4 can be easily interpreted as an El-Niño like pattern, the interpretation of archetype #3 is more ambiguous, suggesting a role for local dynamics not identified by the large-scale patterns extracted by AA”. The explanation is still largely based on pattern match and the description of the missing local dynamics is lacking.

Response to Reviewer Comments on *Chapman et al*: Large-Scale Drivers of Marine Heatwaves Revealed by Archetype Analysis

Christopher Chapman, Dider Monselesan, James Risbey,
Ming Feng and Bernadette Sloyan

July 2022

We would like to sincerely thank all reviewers for taking the time to review our paper and provide such constructive comments. We have made a sincere effort to address them, which has dramatically improved the paper.

Both reviewers note the description of physical mechanisms leading to the onset of marine extremes is somewhat superficial, particularly those related to the temporal evolution. We have attempted to address these comments while adhering to the strict word limit of the journal by incorporating a discussion of mechanisms into both the introduction and conclusion sections of the main text, and clarifying wording within the case studies. However, due to the journal's word limit, a full discussion of causal mechanisms is beyond the scope of the present work. Follow-up work is planned.

Below, we present our response to the each reviewer comment. Original reviewer comments are typeset in *italics*, our responses are in normal text and direct quotes and excerpts from the revised manuscript are typeset in blue.

Response to Reviewer #1

:

Our thanks go to the reviewer for their constructive and supportive comments.

Major Comments

- *Some discussion, on the order of 3-4 paragraphs, should be added to the Discussion at the end regarding the application of AA in light of the temporal nature of teleconnections. It appears that AA works at establishing the connection between large scale Ocean-Atmosphere patterns at zero lag to the regional Marine Heatwave structure. As described in the text in lines 212–213 there are propagation time scales to*

consider. The lagged correlation relating to the climate indices was useful. It would be good to have a more general discussion of how AA relates to lags. I think it implies zero lag only, but it would be good to state this explicitly in the discussion. I only make this point as the AA methodology appears very promising and I expect this to be used in many other geographic region by less skilled researchers, and a clear directive in the discussion would be useful to future efforts.

This is a fair point, and robust understanding of the temporal “propagation” of AA derived patterns is currently an area of active research within our group. In short, AA can not capture serial correlation or temporal patterns in data. If one were to randomly shuffle the time index of the data matrix, the resulting AA spatial patterns would be identical to those obtained from the original data matrix, and both the C and S matrix time-series would simply be re-ordered following the shuffling of the data matrix (and we note that many popular matrix factorisation methods, such as Principal Component Analysis/Empirical Orthogonal Functions, behave in the same way). In a recently published technical manuscript (Black et al. 2022) we have explored some of these issues. However, assessment of temporal characteristics of the various archetypes, using current methodology, requires analysis of the affiliation time-series, encoded in the S -matrix. As the reviewer correctly notes, the S -matrix *applies at zero-lag*. However, time-shifted composites (sometimes called *lagged composites*) can be employed to investigate the temporal propagation of spatial features, as well as investigate related questions such as growth and decay time-scales, persistence, etc...

In response to this comment, we have calculated time-shifted (lagged) composites of SST for all archetypes, which have revealed typical temporal patterns of growth and decay as opposed to obvious spatio-temporal propagation. Additionally, different archetypes show differing time-scales, with some (such as those associated with El-Niño/La-Niña like conditions) being strongly persistent, while others (such as the archetype used in the New Zealand marine-heatwave case study) more ephemeral. We have added the lagged composite maps and a discussion of their time-scales to the Supplementary Material, and integrated a discussion of their time-scale into the individual case studies. We have also added a paragraph to the methods section:

Temporal Patterns and Archetype Analysis

Native implementation of the AA algorithm as described by Eqn. 6 is incapable of directly extracting temporal patterns, such as serial correlation or persistence, from data (Black et al. 2022). For example, a re-ordering of the time index of the data matrix, represented by the operation $\mathbf{X}' = \mathbf{R}\mathbf{X}$ will result in reordered but otherwise identical affiliation and mixture weight

matrices, given by $\mathbf{S}' = \mathbf{S}\mathbf{R}^T$ and $\mathbf{C}' = \mathbf{R}\mathbf{C}$. Although options for including temporal patterns directly into the AA procedure have been discussed (Black et al. 2022), at present, it is only possible to extract through interrogation of the resultant affiliation time-series from the S matrix.

as well as an explanation of the procedure for forming time-shifted composites: At any given time step in the data, t , the value of the affiliation time-series for archetype p , written $S(p, t)$, expresses an estimate of the strength of the archetype's expression *at that timestep with 0 lag*. However, by shifting the time-index of the affiliation probability with respect to the data to form *lagged composites*, which can be used to investigate the temporal propagation of certain spatial features and (equivalently) their associated timescales. We have alluded to the temporal evolution in each of the case studies, and present the results of lagged composite analysis in the supplementary material.

For the Reviewer's information, we reproduce the lagged SST (composited on the affiliation time series) at each of the representative locations used in the case-studies below. It should be clear that in certain regions, the archetype reaches its maximum expression near lag 0 (both the SE Indian Ocean case study and the Tasman Sea case study), while there is evidence that the expression of archetypes #3 and #4 may peak in the GBR region some time following the peak of of the affiliation time-series.

We have attempted to fold some of this discussion into the main text (and

- *My other comments are minor and are listed below. A further check of spelling and grammar would be helpful, not sure I caught everything.*

We thanks the reviewer for their attention to detail and keen eye. We have carefully proof-read the document and hopefully caught all the typos.

- *Line 129. What is dominant in quantitative terms? δ 0.5? Define.*

Yes. We have updated the text.

- *Line 139 el Nino are δ el Nino*

Fixed

- *Line 151 average was the δ average. It was the (split into two sentences)*

Sentence removed in editing.

- *Line 153 that 3C δ than 3C*

Fixed.

- *Line 223 Pacific the surface air δ Pacific surface air*

Sentence removed in editing.

- *Line 266 remove in Figs to both the*

Fixed

Figure 1: Lagged SST for each archetypal pattern at for each main text case study **a** South-east Indian Ocean; **b** South-west Pacific/New Zealand; **c** Great Barrier Reef region.

- *Fig. 5,7 caption marine heatwaves occurrences & marine heatwave occurrences*

Fixed

- *Line 329 studys & study*

Fixed

- *Line 356 run on sentence split into two*

Fixed

- *Line 367 shows & show*

Fixed

- *Line 369 temperatures through & temperatures occur through*

Fixed

- *Line 369 temperatures through & temperatures occur through*

Sentence rephrased in editing.

- *Line 378 remove to have*

Sentence removed in editing.

- *Line 379 month & months*

Sentence removed in editing.

- *Line 387 near GBR & near the GBR*

Sentence removed in editing.

- *Fig. 7 caption 2nd line from bottom marine heatwaves occurrences & marine heatwave occurrences*

Fixed

- *Line 402 scale & scales*

Fixed

- *Line 453 closey & closely*

Fixed

- *Line 458 with extreme the & with extreme*

Fixed

- *Line 463 insteade & instead*

Fixed

- *Line 496 Why However? Doesn't this point just agree with the first point in the first sentence?*

We've removed this sentence because, as the Reviewer notes, it's redundant and we are attempting to consolidate the text to meet the word limits.

- *Line 503 remove Extremes*

Done.

- *Line 584 we ÷ We*

Sentence rephrased

- *Line 665 as an ÷ an*
- *Line 783 that ÷ than*

Fixed

Response to Reviewer #2

Our sincere thanks to the reviewer for their constructive comments.

- *I do, however, have comments on the discussion of the processes and mechanisms about the drivers of the events. My impression is that this work is more like 'proof-of-concept' type of study, rather than revealing the actual drivers of the heatwaves. The description of the mechanisms are mostly brief, vague and sometimes speculative, which does not provide much in-depth knowledge of the driving processes. Since the title of this manuscript contains "Drivers", I suggest that the authors expand the explanations and descriptions of the mechanisms driving the events discussed in the manuscript. Some examples can be found below.*

We agree somewhat with the characterisation of our paper as leaning towards the "proof-of-concept" end of the spectrum. The principle goal of this paper was to illustrate a new method and (perhaps grandiosely) a new way of thinking about the large-scale drivers of extreme events (which we've referred to as "outside-in").

We have attempted to expand the discussion to physical phenomena by drawing more extensively on the literature while keeping within the strict word limit imposed by the journal. In other parts of the manuscript, we have attempted to tighten the language used to be less speculative. This includes a minor change to the title of the manuscript, which is now **A Large-Scale View of Marine Heatwaves Revealed by Archetype Analysis**, that we feel more accurately describes the content.

Particular comments

- Page 10, line 211: “The thermocline anomalies in the western Pacific are known to propagate via oceanic teleconnection to instigate the southeast Indian Ocean marine heatwave.” The authors could have explained more how the oceanic teleconnection works, instead of just inserting a reference.

We’ve expanded the discussion here:

These conditions are known to initiate anomalously strong low-level trade winds in the tropical region, which in-turn result in a transport of warm water from the Pacific to the Indian basins via the Indonesian straits, with the highest temperatures in the south-east Indian Ocean occurring approximately 3-5 months after the most intense *negative* anomalies in the equatorial Pacific.

- Page 10, line 243: “The dominant driver is found to be central Pacific La Niña, with the SAM potential secondary driver”. Again, how does the process work? Also, be careful about whether or not La Nina (SST anomaly) is a driver or response of the actual driver. Similarly, how does SAM (wind variability) drive the SST anomaly in the domain of interest needs more than visual inspection of the patterns.

We agree with the reviewer that the link between the traditional climate modes (SAM and La-Niña) was rather speculative, and in response, we substantially modified this section. The modified description of the influence of the atmosphere is now:

The anomalous atmospheric circulation associated with this archetype (Fig. 2b) shows both local and remote anomalies. Locally, a cyclonic mid-tropospheric circulation anomalies directs airflow from the north and east, which are likely to contribute to the above average surface temperatures by bringing warmer continental and tropical air to the region. In the equatorial Pacific, the the mid-tropospheric winds show the characteristic divergence pattern associated with the central Pacific La Niñas. A ridge of high pressure extends across the Southern Ocean and anomalously strong eastward winds associated with the polar jet stream are found south of 55°S, which is a feature characteristic of the positive phase of the Southern Annual Mode (SAM).

While the concluding paragraph to this section has also been modified:

Previous studies of the extreme 2010-2011 south-east Indian Ocean marine heatwave attribute $\sim 2/3$ s of the excess warming to anomalous ocean heat transport, due to remote conditions in the equatorial Pacific, while the remaining $\sim 1/3$ due to enhanced surface heating driven by local atmospheric processes. It is notable that the AA largely confirms these patterns, suggesting remote influences from the tropical Pacific, reminiscent of central Pacific La-Niña, with a time lag of ~ 5 months, and local influences from a stationary mid-tropospheric cyclone. Our analysis indicates that the large-scale

conditions responsible for the extreme 2010-2011 event are recurring and could form the basic ingredients of an “extreme climate mode” that strongly influences the south-east Indian ocean.

- *Page 11, 294: “The spatial patterns shown in Figs. 6a,b suggest that marine heatwaves around New Zealand are associated with classical La Niña type patterns, as well persistent atmospheric blocking high pressure systems.” This sentence seems to suggest that every heatwave around New Zealand is associated with La Nina and blocking. Is this the case?*

We have modified the text here (and elsewhere in the section, following the reviewer’s suggestion to avoid speculative links between the archetype and various climate drivers):

Our analysis suggests that localised atmospheric blocking may be a strong driver of the most extreme of persistent marine heatwaves in the southern Tasman sea, and role of broad-scale teleconnections is uncertain. The similarity of the archetypal SST patterns to that of the composite average and the clustering of events during periods when the best-matching archetype is strongly expressed also suggests these patterns are reoccurring and associated with most (although, notably, not all) marine heatwaves in the region. Blocking highs have long been associated with extreme weather events, including marine heatwaves, and there is currently no generally accepted theory that completely explains their dynamics. Certain persistent atmospheric regimes, such as blocking, can be detected using AA, and future work could seek to integrate these analyses to improve understanding of the dynamics of these events.

- *Page 14, line 318: “The analysis conducted here suggests that although both classical La-Niña and PSA climate modes play a role on driving marine heatwaves around New Zealand, individually each of these climate drivers has only a weak influence, with the dominant role being played by localised atmospheric blocking high pressure systems.” Again, description of the driving process is inadequate.*

We have modified the text here (see response to the comment above).

- *Page 15, line 349: “Unlike in the previous case studies, we find that at least 2 archetypes, archetypes #3 (Fig 7c) and #4 (Fig. 7e) are required to ...”. The determination of which archetypes explain the heatwave seems arbitrary. Is there a more quantitative way of selecting the archetypes?*

Although we disagree with the term “arbitrary”, although we note that the selection of the best matching archetype is not necessarily “objective”. We now include a more detailed discussion of how we select the best matching archetype in the methods section:

Selection of the ‘best matching archetype’

Determination of the best matching archetype for the case studies presented is performed manually using semi-objective criteria. However, there is some subjectivity and a truly objective method.

First, the spatial SST pattern of each archetype is assessed at each representative location by linearly interpolating of the fields shown in Fig. 1. Only those archetypes with a strong expression at each representative location were considered. Then, the spatial patterns of the archetype are examined for similarity with the composite average of all marine heatwave events detected at that location (as in Figs. 3,5 and 7), and the affiliation time series for that archetype examined for its similarity with the SST anomalies and the temporal distribution of marine heatwave events.

In the first two case studies presented here (as well as those in the supplementary material) the best-matching archetype was relatively obvious based on spatial patterns alone. However, the Coral Sea case study required some care in selecting archetypes, as only summertime (December, January, February) events were considered (as these events lead to coral bleaching) and no one archetype was consistently consistently associated with summertime marine heatwaves. As such, we selected the only two archetypes that had positive expressions at the representative location during the appropriate season. Further detail is included in the supplementary material.

- *Page 19, line 393: “While archetype #4 can be easily interpreted as an El-Niño like pattern, the interpretation of archetype #3 is more ambiguous, suggesting a role for local dynamics not identified by the large-scale patterns extracted by AA”. The explanation is still largely based on pattern match and the description of the missing local dynamics is lacking.*

This is a fair comment and in response we have performed a reevaluation of the GBR case study. Previously, we sought a clear link between the tropical Pacific climate mode (El-Niño) present in both archetypes. While the connection to marine heatwaves in the GBR region is relatively clear in the case of the classical El-Niños (archetype #4), the events associated with archetype #3 are somewhat more difficult to interpret dynamically. After several discussions with colleagues with expertise in the GBR, we have found what we think is a plausible dynamical explanation for the co-occurrence of marine heatwaves/bleaching and archetype #3. As a result, we have extensively edited this section, including a more detailed discussion of dynamics (although the word limit naturally constrains the level of detail and analysis). We won’t copy-paste the revisions here, as they are quite lengthy, but we do draw the Reviewer’s attention to the revised section.

REVIEWERS' COMMENTS

Reviewer #1 (Remarks to the Author):

This manuscript has had extensive revisions that clarify the key point about the relation to temporal variability and teleconnections. The dynamics of the events are not within the scope of the work, but helpful comments on interpretation of atmospheric inputs likely forcing the Marine Heatwaves have been added. The issues raised by both reviewers have been addressed in detail. I find this manuscript to be acceptable in its present form and commend the authors for doing such a thorough job with revisions. There are still numerous small issues with the writing, below are some detailed comments.

Line 17. Large scale > large scale patterns (?)

Line 41 in in > in

Line 65 heatwaves > heatwave

Line 79. Delete and describes

Line 119. Through Great > through the Great

Line 128. As 16 > as a 16

Line 129 2012 period > 2012

Line 133. Revealing > reveals

Figure 2 caption. "In figure text" is confusing

Figure 3 caption line 4. Heatwave > heatwaves

Line 177 in > is

Line 188. The accompany > that accompany

Line 209 a anomalous > an anomalous

Line 230 1/3 due > 1/3 is due

Line 279 add space between comma and "constant"

Line 283 well persistent > well as persistent

Line 292 reminiscnet > reminiscent

Line 299 and role > and the role

Line 337. Appear co-occur > appear to co-occur

Line 353 delete "are weak"

Line 545 principle > principal

Response to Reviewer Comments on *Chapman et al*: Large-Scale Drivers of Marine Heatwaves Revealed by Archetype Analysis

Christopher Chapman, Dider Monselesan, James Risbey,
Ming Feng and Bernadette Sloyan

July 2022

We would like to sincerely thank all reviewers for taking the time to review our paper and for providing constructive comments across the two rounds of review. We are pleased that we have been able to largely meet their concerns.

Response to Reviewer #1

:

Our thanks go to the reviewer for their constructive and supportive comments.

Major Comments

- *There are still numerous small issues with the writing, below are some detailed comments.*

We thanks the reviewer for their attention to detail. We hope that we have caught all typos now.

- *Large scale $\dot{\}$ large scale patterns (?)*

Yes. Fixed.

- *Line 41 in in $\dot{\}$ in*

Fixed

- *Line 65 heatwaves $\dot{\}$ heatwave*

Fixed

- *Line 79. Delete “and describes”*

Fixed.

- *Line 119. Through Great δ through the Great*

Fixed.

- *Line 128. As 16 δ as a 16*

Fixed

- *Line 129 2012 period δ 2012*

Fixed

- *Line 133. Revealing δ reveals*

Fixed

- *Figure 2 caption. "In figure text" is confusing*

Changed to "annotations"

- *Figure 3 caption line 4. Heatwave δ heatwaves*

Fixed

- *Line 177 in δ is*

We cannot locate this error.

- *Line 188: The accompany δ that accompany*

Fixed

- *Line 230: 1/3 due δ 1/3 is due*

Fixed

- *Line 279 add space between comma and "consistent"*

Done.

- *Line 283 well persistent δ well as persistent*

Fixed.

- *Line 292 reminiscnet δ reminiscent*

Fixed

- *Line 299 and role δ and the role*

Fixed

- *Line 337. Appear co-occur δ appear to co-occur*

Fixed

- *Line 353 delete "are weak"*

Fixed

- *Line 545 principle δ principal*

Fixed